# STEPWISE FEATURE LEARNING IN SELF-SUPERVISED LEARNING

## ABSTRACT

Recent advances in self-supervised learning (SSL) have shown remarkable progress in representation learning. However, SSL models often exhibit shortcut learning phenomenon, where they exploit dataset-specific biases rather than learning generalizable features, sometimes leading to severe over-optimization on particular datasets. We present a theoretical framework that analyzes this shortcut learning phenomenon through the lens of *extent bias* and *amplitude bias*. By investigating the relations among extent bias, amplitude bias, and learning priorities in SSL, we demonstrate that learning dynamics is fundamentally governed by the dimensional properties and amplitude of features rather than their semantic importance. Our analysis reveals how the eigenvalues of the feature cross-correlation matrix influence which features are learned earlier, providing insights into why models preferentially learn shortcut features over more generalizable features.

## 1 INTRODUCTION

While deep neural networks have shown remarkable success in various learning tasks, recent studies have revealed a concerning trend: models often exhibit unexpected learning behavior, particularly shortcut learning, which tends to take easier but potentially less reliable paths to solve general tasks (Geirhos et al., 2020). For example, in image classification tasks, models tend to learn earlier larger background features than smaller foreground objects (Hermann et al., 2023), potentially leading them to classify cows based on whether they appear on grass rather than learning actual cow features, or identify camels primarily by detecting desert backgrounds (Beery et al., 2018). This phenomenon is prevalent even in SSL (Doersch et al., 2015b; Noroozi et al., 2017; Wei et al., 2018; Doersch et al., 2015a).

While previous research has shown that neural networks are vulnerable to spurious correlations in data (Arjovsky et al., 2019), several other contributing factors to shortcut learning have been identified. Hermann et al. (2023) find shortcuts emerging from color, size, and background. Rahaman et al. (2019); Tancik et al. (2020) find spectral bias that low-frequency features are learned faster than high-frequency features. While significant progress has been achieved, current theoretical frameworks provide insufficient explanations for why models consistently take shortcuts.

Recent studies have demonstrated that SSL models with small weight initialization exhibit stepwise learning dynamics, where features are learned sequentially based on the corresponding eigenvalues of the feature cross-correlation matrix (Simon et al., 2023). Building on this insight, we analyze the eigenvalue and eigenvector structure of the feature cross-correlation matrix. This approach provides a novel theoretical framework for understanding why certain features, regardless of their semantic importance, are consistently learned earlier in the training process. Our investigation focuses particularly on how dimensional properties influence learning priority, potentially explaining some observed shortcut learning phenomena beyond traditional spurious correlations.

The contributions of our work are as follows:

- We establish theoretical connections between shortcut learning phenomena, stepwise learning, and eigenvalue-eigenvector of feature cross-correlation matrix on SSL.

- We extend theoretical research on shortcut learning from supervised learning to SSL.

- We characterize *extent bias*, a tendency to prioritize features based on their dimensional extent or spatial coverage rather than their semantic importance.
- We analyze how amplitude and frequency determine which features are learned earlier in SSL, and characterize *amplitude bias*, a tendency to prioritize features based on their amplitude rather than their semantic importance.

## 2 RELATED WORKS

**Self-supervised learning**   Early contrastive methods like SimCLR (Chen et al., 2020) required large batches, motivating non-contrastive approaches like SimSiam (Chen & He, 2021) and BYOL (Grill et al., 2020). Subsequent methods introduced novel regularizers, such as the variance-invariance-covariance in VICReg (Bardes et al., 2021) and the cross-correlation matrix in Barlow Twins (Zbontar et al., 2021), to prevent representational collapse. DINO (Caron et al., 2021) advanced the field by introducing self-distillation with no labels. The success of DINO v2 (Oquab et al., 2023) sparked interest in Joint Embedding Predictive Architectures (JEPA) (Assran et al., 2023), with recent work by Littwin et al. (2024) revealing JEPA's tendency to prioritize learning "related" features over "frequently" occurring ones.

**Learning dynamics**   Following the introduction of Neural Tangent Kernel (NTK) (Jacot et al., 2018), researchers have discovered important connections between eigenvalue dynamics and learning behavior, including spectral bias phenomena (Tancik et al., 2020; Halvagal et al., 2022). This theoretical framework has enabled deeper analysis of loss function trajectories and saddle point behaviors (Jacot et al., 2021; Pesme & Flammarion, 2023). Notably, Simon et al. (2023) demonstrated that these saddle-to-saddle dynamics appear not only in supervised learning but also extend to SSL settings.

**Shortcut learning**   Shortcut learning was first identified in Geirhos et al. (2020), describing how neural networks take easier but incorrect paths to solve tasks. This phenomenon appears in various ways: Geirhos et al. (2018); Baker et al. (2018); Hermann & Lampinen (2020) showed that CNNs rely on object texture rather than object shape, Wu et al. (2022) demonstrated that even a single pixel can mislead model's decisions, and Hermann et al. (2023) revealed that CNNs preferentially learn salient but potentially irrelevant features like scale and background elements. These shortcuts can arise from dataset properties, particularly through spurious correlations (Arjovsky et al., 2019) and implicit biases. Our work specifically examines how dataset correlations contribute to shortcut learning.

## 3 BACKGROUND (STEPWISE NATURE OF SSL (SIMON ET AL., 2023))

In this section, following Simon et al. (2023), we analyze the stepwise learning dynamics of SSL systems through the lens of toy Barlow Twins models (Zbontar et al., 2021). We first introduce the loss function and gradient flow dynamics, then derive the connection between cross-correlation matrix and feature learning. Finally, we examine how the eigendecomposition of feature cross-correlation matrix connects to the theoretical foundation for our analysis of extent bias and amplitude bias.

Given training data $\{x^{(i)} \in \mathbb{R}^m : i = 1, 2, \cdots, n\}$, the training loss of toy Barlow Twins is defined as $\mathcal{L} = ||C - I_d||_F^2$, where $|| \cdot ||_F$ is Frobenius norm, $C \equiv \frac{1}{2n} \sum_{i=1}^n (Wx^{(i)})(Wx'^{(i)})^\top + (Wx'^{(i)})(Wx^{(i)})^\top \in \mathbb{R}^{d \times d}$ is cross-correlation matrix of $Wx$ and $Wx'$ for augmented view $x'$ from $x$, and $W \in \mathbb{R}^{d \times m}$ represents learnable parameters. Using the feature cross-correlation matrix

$$\Gamma \equiv \frac{1}{2n} \sum_{i=1}^n (x^{(i)} x'^{(i)\top} + x'^{(i)} x^{(i)\top}) \in \mathbb{R}^{m \times m}, \tag{1}$$

we have

$$\mathcal{L} = ||W\Gamma W^\top - I_d||_F^2, \; C = W\Gamma W^\top. \tag{2}$$

The eigendecomposition of the feature cross-correlation matrix is $\Gamma = V_\Gamma \Lambda_\Gamma V_\Gamma^\top$ with $\Lambda_\Gamma = \text{diag}(\gamma_1, \cdots, \gamma_m)$ and $V_\Gamma = [v_1 \; \cdots \; v_m] \in \mathbb{R}^{m \times m}$, where $\gamma_1 \geq \gamma_2 \geq \cdots \geq \gamma_m$ are eigenvalues of $\Gamma$ and $v_i$'s are the corresponding eigenvectors for $\gamma_i$'s.

To analyze the eigenvector dynamics of the weights, we assume weight initialization is aligned.

**Assumption 1** (Aligned Initialization (Simon et al., 2023)). At the initialization, we assume that the right-singular vectors of $W(0)$ are aligned with the top $d$ eigenvectors of $\Gamma$, i.e., the singular value decomposition is $W(0) = US_0 V_\Gamma^{(\leq d)\top}$ for a orthogonal matrix $U \in \mathbb{R}^{d \times d}$, the top-$d$ eigenvector matrix $V_\Gamma^{(\leq d)} = [v_1 \cdots v_d] \in \mathbb{R}^{m \times d}$, and a diagonal matrix $S_0 = \text{diag}(s_1(0), \cdots, s_d(0))$ with a small initialization $s_j(0) > 0$.

Under Assumption 1, the solution $W(t)$ for gradient flow $\frac{dW}{dt} = -\nabla_W \mathcal{L} = -4(W\Gamma W^\top - I_d)W\Gamma$ can be expressed as follows (Simon et al., 2023, Proposition 4.1): $W(t) = US(t)V_\Gamma^{(\leq d)\top}$ for $S(t) = \text{diag}(s_1(t), \cdots, s_d(t))$, where the singular values of $W(t)$ evolve as $s_j(t) = \frac{e^{4\gamma_j t}}{\sqrt{s_j^{-2}(0) + (e^{8\gamma_j t} - 1)\gamma_j}}$ which has a limit of $\gamma_j^{-1/2}$ as $t \to \infty$ and nearly sigmoidal

$$s_j^2(t) \approx \frac{1}{\gamma_j + s_j^{-2}(0)e^{-8\gamma_j t}} =: \tilde{s}_j^2(t). \tag{3}$$

Solving $\tilde{s}_j^2(t) = \frac{1}{2}s_j^2(\infty)$ at its critical time $t = \tau_j$, we have

$$\tau_j = -\log\left(s_j^2(0)\gamma_j\right)/8\gamma_j \tag{4}$$

around which $s_j(t)$ (or $\tilde{s}_j(t)$) passes $\frac{1}{2}\gamma_j^{-1/2}$ and rapidly increases from near zero to near the saturation $\gamma_j^{-1/2}$.

In this paper, we focus on the property that the eigenvector feature $v_j$ corresponding to a larger $\gamma_j$ leads to an earlier critical point $\tau_j$ from (4).

## 4 EXTENT BIAS

In computer vision tasks, backgrounds typically span larger regions while foreground objects occupy more concentrated areas. Recent work by Hermann et al. (2023) reveals that CNNs preferentially learn these background features over object-specific details, creating a specific form of spurious correlation between backgrounds and class labels. For example, cows are often classified based on grass backgrounds rather than their distinctive features, and camels are identified through desert scenes (Beery et al., 2018).

**Distinguishing extent and amplitude bias** This phenomenon points to an underlying learning mechanism we term *extent bias*. To fully characterize this mechanism, it is crucial to clarify that our framework investigates two distinct and naturally separable mechanisms that influence feature learning priority: *extent bias*—the tendency to prioritize features based on their spatial coverage (how many dimensions/pixels a feature spans), and *amplitude bias*—the tendency to prioritize features based on their per-coordinate intensity, independent of spatial coverage. These represent orthogonal factors:

- **Extent bias** (this section): Features differ in spatial coverage while maintaining fixed per-coordinate amplitude.
- **Amplitude bias** (Section 5): Features occupy identical spatial coverage while varying in per-coordinate amplitude.

The connection between extent bias and learning dynamics highlights the need to understand a more fundamental mechanism beyond traditional spurious correlations.

While spurious correlations emerge from dataset-specific relationships, the bias toward learning background features is inherent in the learning dynamics of neural networks themselves. Through our analysis of SSL systems, we demonstrate that this bias for background features emerges naturally from how models learn earlier features with higher extent bias, independent of their semantic relevance or predictive power.

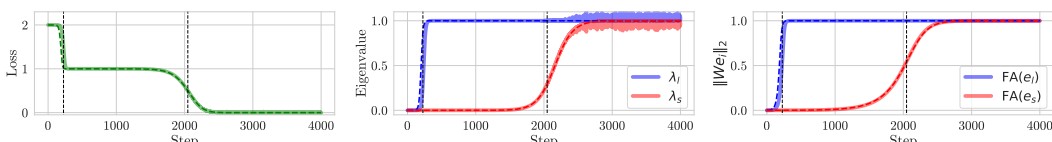

Figure 1: **Effects of extent bias on learning dynamics in SSL.** (Left) Stepwise learning curves of Barlow Twins. There are two ($d = 2$) learning steps shown with two black dashed vertical lines (also shown in the other two panels) which indicate the time steps $t_1$ and $t_2$ with $t_1 : t_2 \approx \frac{1}{\gamma_l} : \frac{1}{\gamma_s} = \frac{1}{m_l} : \frac{1}{m_s}$. The predicted loss (dashed green) of $\mathcal{L} = \sum_{j=1}^d (\tilde{\lambda}_j(t) - 1)^2 = \sum_{j=1}^d (\tilde{s}_j^2(t)\gamma_j - 1)^2$ using (3) match the empirical result (solid green). (Center) Evolution of eigenvalues $\lambda_j$'s of $C$ during training. At the beginning, the first eigenvalue $\lambda_1$ (blue) increases to 1 and then later the second $\lambda_2$ (red) follows. We also compare them with the predicted evolution $\tilde{\lambda}_j(t)$ (dashed lines). (Right) Evolution of the feature alignment $\|We\|_2$ for $e = e_l$ (blue) and $e = e_s$ (red). It shows very similar behaviors with the eigenvalues $\tilde{\lambda}_j^{1/2}$ (dashed lines). See Theorem B.1. We use $m_l = 9$, $m_s = 1$. See Section D.1 for more detailed settings.

In this section, we investigate how different feature properties influence learning priorities in SSL. Through extent bias analysis, we demonstrate how features with larger dimensional coverage are learned before those with smaller coverage, regardless of their semantic importance.

We construct a theoretical framework that identifies dimensional effects in feature learning. By analyzing how SSL models process features of varying extent bias, we can directly observe how extent bias influences learning priority and connects to the background-foreground learning dynamics observed in practice.

### 4.1 SETTINGS

We first consider the following base input $x_{\text{base}} = [b_l \mathbf{1}_{m_l}^\top, b_s \mathbf{1}_{m_s}^\top]^\top \in \mathbb{R}^m$, where $b_l, b_s \stackrel{\text{i.i.d.}}{\sim} B(p = 0.5)$ follow the Bernoulli distribution and take the value $\pm 1$ with the equal probability, $m_l$ and $m_s$ indicate the size of larger part and smaller part, respectively, i.e., $m_l > m_s$ and $m_l + m_s = m$, and $\mathbf{1}_k$ is the $k$-dimensional all-one vector. From now on, we will use the subscript $l$ and $s$ for the indices with respect to the *larger*-part and *smaller*-part features, respectively.

Then, to obtain the augmented pairs $(x, x')$, we introduce the following data augmentation $x = x_{\text{base}} + \varepsilon$ and $x' = x_{\text{base}} + \varepsilon'$, with the noise $\varepsilon, \varepsilon' \stackrel{\text{i.i.d.}}{\sim} \mathcal{N}(0_m, a^2 I_m)$ for some $a > 0$.

### 4.2 LEARNING DYNAMICS ON EXTENT BIAS

In this subsection, we discuss the relationship between $\gamma_j$ and $\mathcal{L}$, focusing on which features are learned earlier. From Section 4.1, we can simplify the feature cross-correlation matrix $\Gamma$ by analyzing the expected values of the augmented features. Based on the definition in (1), we have:

$$\Gamma = \frac{1}{2n} \sum_{i=1}^n (x^{(i)} x'^{(i)\top} + x'^{(i)} x^{(i)\top}) = \mathbb{E}[x_{\text{base}} x_{\text{base}}^\top]. \tag{5}$$

To identify which features drive the loss as stepwise phenomena, we consider basis vectors that disentangle individual features. Specifically, we define basis vectors $e_l$ and $e_s$ where each vector has ones only in the dimensions corresponding to its respective feature:

$$e_l = [\mathbf{1}_{m_l}^\top, \mathbf{0}_{m_s}^\top]^\top, e_s = [\mathbf{0}_{m_l}^\top, \mathbf{1}_{m_s}^\top]^\top \in \mathbb{R}^m.$$

By measuring the feature alignment $\text{FA}(e) = \|We\|_2$ between the weight matrix and the basis vectors $e = e_l, e_s$, we can identify which features are being learned at each stage of the training process.

The eigendecomposition of $\Gamma$ is given by the following proposition:

**Theorem 4.1.** *The correlation matrix in (5) has the eigenvalue matrix $\Lambda_\Gamma$ and eigenvector matrix $V_\Gamma$:*

$$\Lambda_\Gamma = diag\left([m_l, m_s, \ \mathbf{0}_{m-2}]\right), V_\Gamma^{(\leq 2)} = [e_l/\sqrt{m_l} \ e_s/\sqrt{m_s}].$$

We defer the proof to Section A.1.

We hypothesize that features with larger dimensions are learned faster, regardless of their predictive power or potential to cause shortcuts. This is particularly relevant in vision tasks where such features might correspond to larger pixel regions. We experiment using a simple toy model to validate our theoretical analysis of dimensional influence on feature learning. In our experimental setup, we used two distinct features with different dimensional coverage ($m_l = 9$ and $m_s = 1$), allowing us to clearly observe the learning dynamics.

As shown in Figure 1, the model's stepwise learning dynamics are governed by the eigenvalues of the feature cross-correlation matrix, resulting in distinct loss drops. The evolution of eigenvalues (Center) and the feature alignments (Right) provide direct evidence of the learning order determined by the eigenvalue dynamics that the alignment with the higher-dimensional feature, $e_l$, increases first and the alignment with $e_s$ follows. This result suggests that the spatial extent of features, rather than their semantic content, plays a crucial role in determining learning priority.

### 4.3 CROSS-CORRELATION EIGENVALUE AND LOSS RELATIONSHIP

In this subsection, we analyze the relationship between the eigenvalues $\lambda_j$ of cross-correlation matrix.

**Theorem 4.2.** *Under Assumption 1, the eigenvalues $\lambda_j$ of feature cross-correlation matrix $C$, using the approximation $s_j^2 \approx \tilde{s}_j^2$ in (3), are approximated as $\lambda_j = s_j^2 \gamma_j \approx \tilde{s}_j^2 \gamma_j =: \tilde{\lambda}_j$ which have*

$$\tilde{\lambda}_j(\tau_j) = \frac{1}{2} \text{ and } \tilde{\lambda}_i'(\tau_j) \begin{cases} = 2\gamma_j & \text{if } i = j, \\ \approx 0 & \text{if } i \neq j \end{cases} \tag{6}$$

*at $\tau_j = -\log(s_j^2(0)\gamma_j)/8\gamma_j$ in (4). For the Barlow Twins loss $\mathcal{L} = \|C - I_d\|_F^2$, we have $\mathcal{L} = \sum_{j=1}^d (\lambda_j - 1)^2$ and $-\frac{d\mathcal{L}}{dt}(\tau_j) \approx \tilde{\lambda}_j'(\tau_j) = 2\gamma_j$.*

We defer the proof to Section A.3.

Figure 15 in Section B.1.4 shows the relationship between the derivative of cross-correlation eigenvalues $\lambda_j'$ and the loss derivative $-\frac{d\mathcal{L}}{dt}$. The close alignment between the loss derivative and $\lambda_j'$ curves demonstrates that the decrease in loss is directly driven by $\lambda_j$, with larger $m_l$ features learned, and smaller $m_s$ features learned later. The curves' relative magnitudes show an approximate $m_l : m_s$ ratio, which matches our theoretical predictions.

### 4.4 WEIGHT SINGULAR VALUE EVOLUTION

To verify the dynamics of weight singular values $s_j$, we propose the following theorem:

**Theorem 4.3.** *Using the approximation (3), the singular values of the weight matrix $W$ satisfy*

$$\tilde{s}_j(\tau_j) = 1/\sqrt{2\gamma_j} \text{ and } \tilde{s}_j'(\tau_j) = \sqrt{2\gamma_j}$$

*at the critical point $t = \tau_j$.*

We defer the proof to Section A.4.

Figure 16 in Section B.1.4 shows two key aspects of singular value dynamics during training. First, the singular values $s_j$ evolve to their theoretical limits $1/\sqrt{\gamma_j}$ and $1/\sqrt{\gamma_s}$, as predicted by our analysis. Second, the derivatives of these singular values exhibit peaks at their respective critical points, with magnitudes that follow the predicted $\sqrt{2\gamma_l} : \sqrt{2\gamma_s}$ ratio. These results provide strong empirical validation of our theoretical framework, demonstrating that both the convergence values and learning priority on different features are governed by their corresponding eigenvalues in the feature cross-correlation matrix $\Gamma$.

### 4.5 ALIGNED INITIALIZATION AND SUBSPACE ALIGNMENT

While Assumption 1 assumes perfect alignment at initialization, Simon et al. (2023) demonstrate that this assumption can be relaxed significantly. They show that even with generic small random initialization, the dynamics quickly converge to the aligned trajectory. This result significantly

strengthens our analysis by showing that the aligned initialization assumption is not restrictive, any sufficiently small initialization will rapidly align with the top eigenvectors of $\Gamma$ before substantial feature learning begins. To validate this theoretical assumption, we define the subspace alignment metric and measured in Section B.1.1.

## 5 Amplitude Bias

Building on our analysis of extent bias, we study amplitude bias–how feature magnitude affects learning priority. To isolate amplitude effects, we need features with identical spatial coverage but different magnitudes. Sinusoidal functions provide an ideal framework, as frequency components can have identical spatial extent while varying in amplitude through their coefficients, allowing us to disentangle amplitude from dimensional coverage effects. This approach connects with existing frequency-based learning research (Rahaman et al., 2019; Tancik et al., 2020; Wang et al., 2023), which has primarily focused on supervised learning. By studying sinusoidal features in SSL, we investigate whether amplitude or frequency characteristics more strongly determine learning priority while extending frequency-based analysis to self-supervised settings.

### 5.1 Settings

To analyze how frequency and amplitude properties affect learning dynamics, we consider input data $x_{\text{base}} \in \mathbb{R}^m$ composed of two sinusoidal components with different frequencies:

$$x_{\text{base}}[t] = c_{ha} b_{ha} \sin(f_{ha} t) + c_{la} b_{la} \sin(f_{la} t), \tag{7}$$

where $f_{ha} = \frac{2\pi}{m} k$ and $f_{la} = \frac{2\pi}{m} k'$ represent different frequencies for some integers $k$ and $k'$, $b_{ha}, b_{la} \overset{\text{i.i.d.}}{\sim} B(p = 0.5)$ follow the Bernoulli distribution and take the value $\pm 1$. Suppose $f_{ha} < f_{la}$ to examine the learning dynamics between low and high frequency components. The coefficients $c_{ha}$ and $c_{la}$ control the amplitude of each sinusoidal component, allowing us to investigate which features are learned earlier. The Bernoulli variables $b_{ha}$ and $b_{la}$ introduce phase reversal in the signal. The time vector $t$ spans the input dimension $m$. We use the same augmentation with (4.1) to generate augmented pairs $(x, x')$ by adding Gaussian noise.

### 5.2 Learning Dynamics on Amplitude Bias

Similar to Section 4.2, we consider basis vectors $e_{ha}$ and $e_{la}$ that isolate individual features: $e_{ha} = c_{ha} \sin(f_{ha} t)$ and $e_{la} = c_{la} \sin(f_{la} t)$, where $0 \leq t \leq m$. Note that these two are orthogonal since $f_{ha} = \frac{2\pi}{m} k$ and $f_{la} = \frac{2\pi}{m} k'$ with $k \neq k'$. Similar to Theorem 4.1, the cross-correlation matrix $\Gamma$ for the data generated from (7) can be expressed as follows:

**Theorem 5.1.** *Under (7), the correlation matrix $\Gamma$ has*

$$\Lambda_\Gamma = diag\left(\left[c_{ha}^2 m/2, c_{la}^2 m/2, \mathbf{0}_{m-2}\right]\right), V_\Gamma^{(\leq 2)} = [e_{ha}/c_{ha} \ \ e_{la}/c_{la}]. \tag{8}$$

We defer the proof to Section A.2.

From Equation (8), we observe that eigenvalues are proportional to the squares of the coefficients $c_{ha}^2$ and $c_{la}^2$. This implies that the learning dynamics are more strongly influenced by the amplitude rather than the underlying frequency.

To validate our theoretical analysis of amplitude bias effect on learning dynamics, we conduct experiments using input data defined in (7). Especially, we set $c_{ha} > c_{la}$. This configuration shown in Figure 10 in Section D, allows us to examine how high-amplitude $c_{ha} \sin(f_{ha} t)$ and low-amplitude $c_{la} \sin(f_{la} t)$ affects feature amplitude bias. More details about the experiment are in Section B.1.2.

Our analysis reveals two dominant eigenvalues corresponding to high-amplitude and low-amplitude components. The eigenvectors of $\Gamma$ (shown in Figure 11, Section D) capture these oscillations: the first eigenvector corresponds to the dominant high-amplitude oscillation, the second to the low-amplitude oscillation, while other eigenvectors are noise with eigenvalues near zero.

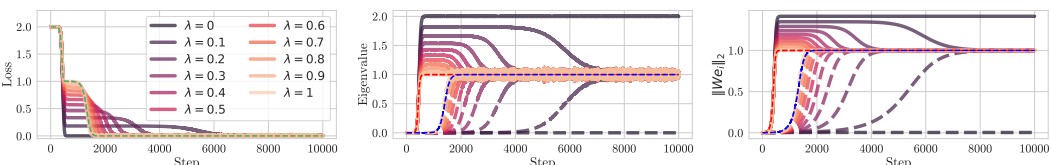

Figure 2: Effects of redundancy reduction coefficient $\lambda$ of the general Barlow Twins loss on linear network dynamics with $m_l = 6, m_s = 2$. (Left) Training loss evolution: empirical results (solid lines) and theoretical prediction for $\lambda = 1$ (green dashed). (Center) Eigenvalue dynamics: solid lines show $\lambda_l$ (larger extent bias) and $\lambda_s$ (smaller extent bias), with red/blue dashed lines as theoretical predictions for $\lambda = 1$. (Right) Feature alignment evolution: solid/dashed lines show FA($e_l$) and FA($e_s$), with red/blue dashed lines as theoretical predictions for $\lambda = 1$.

### 5.3 Discussion

Figure 12 in Section B.1.2 shows that a learning process is driven primarily by feature coefficient magnitude rather than frequency characteristics. The key observation is that the first learned features are those with large coefficients, independent of their spectral properties. This finding parallels frequency shortcut (Wang et al., 2023) in classification tasks, but reveals a different underlying mechanism. While frequency shortcut suggests models preferentially learn distinctive Fourier components, our results demonstrate that amplitude magnitude—not frequency characteristics—primarily determines feature learning priority. More detailed results in Section B.1.2.

## 6 General Settings

### 6.1 Redundancy reduction coefficient $\lambda \neq 1$

**Proposition 1.** *For the general Barlow Twins loss, $L_\lambda = (1 - \lambda)L_0 + \lambda L_1$, the redundancy reduction coefficient $\lambda$ governs the learning dynamics by balancing feature learning ($L_0$) and decorrelation ($L_1$), creating a spectrum of behaviors. At the $\lambda = 0$, only the top eigenvalue is learned ($s_1 \to \sqrt{d/\gamma_1}$). Conversely, at $\lambda = 1$, all features are learned independently ($s_k \to \sqrt{1/\gamma_k}$ for all $k$). For intermediate values, $0 < \lambda < 1$, the dynamics are coupled, where the learning of new features can suppress those previously acquired.*

This analysis reveals that smaller $\lambda$ promotes specialization to dominant features, while larger $\lambda$ encourages learning of the full feature space. The detailed derivations are provided in Section A.6.

### 6.2 Deep Linear Networks (DLN)

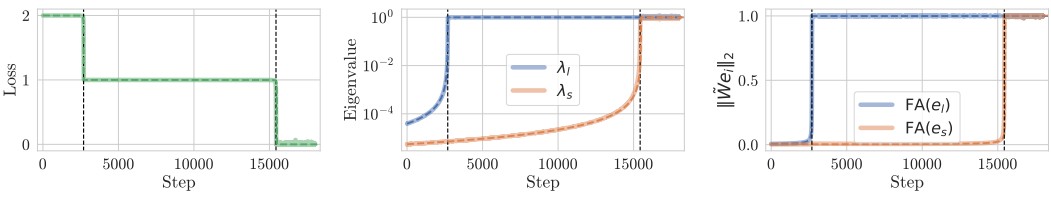

Figure 3: Ideal initialization condition in DLN, assumed in Section A.5. (Left) Stepwise learning curves of Barlow Twins. The predicted loss (dashed green) of $\mathcal{L} = \sum_{j=1}^{d} (\tilde{\lambda}_j(t) - 1)^2 = \sum_{j=1}^{d} (\tilde{s}_j^2(t)\gamma_j - 1)^2$ using (3) match the empirical result by approximation of differential equations (9) (solid green) (Center) Evolution of eigenvalues $\lambda_j$'s of $C$ during training. We compare them with the predicted evolution $\tilde{\lambda}_j(t)$ (dashed lines). (Right) Evolution of the feature alignment $||We||_2$ for $e = e_l$ (blue) and $e = e_s$ (red). It shows very similar behaviors to the eigenvalues $\tilde{\lambda}_j^{1/2}$ (dashed lines).

While we could analyze deep neural networks based on kernels as in our foundational work (Simon et al., 2023), kernel-based analysis has fundamental limitations in explaining feature learning dy-

namics. Even NTK analysis (Woodworth et al., 2020; Nam et al., 2025) cannot capture the feature learning that occurs rich regime. Although DLNs are linear networks, their learning dynamics is non-linear due to multiplicative weight interactions between layers. This enables feature learning while maintaining analytical tractability that kernel methods cannot provide (Ziyin et al., 2022). Similar to recent analysis of Littwin et al. (2024), we provide our analysis in Section A.5.

### 6.3 Non-linear Models (Leaky ReLU, Batch Normalization, and SimCLR)

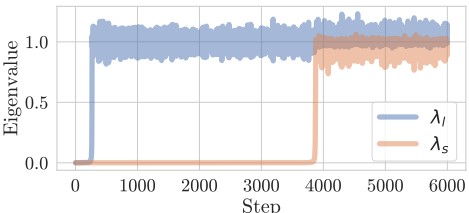

(a) **MLP with ReLU activation.** See Section B.3 for more detailed settings.

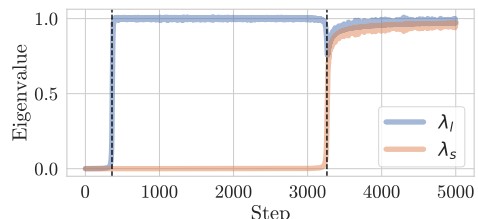

(b) **DLN with BatchNorm ($\lambda = 1$).** See Section E.4 for more detailed settings.

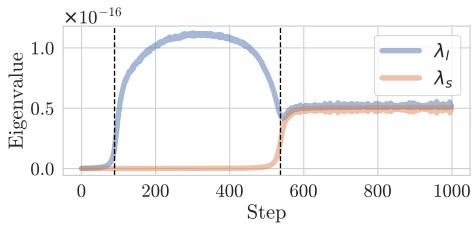

(c) **DLN with SimCLR.** See Section C.1 for more detailed settings.

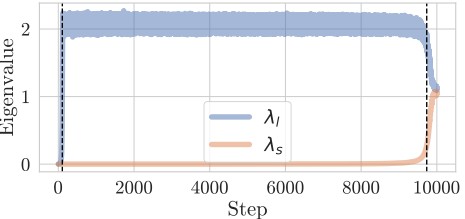

(d) **DLN with VICReg.** See Section C.2 for more detailed settings.

Figure 4: Eigenvalue evolution for non-linear models.

Figure 4 provides a more rigorous verification with the non-linear models (see Section E.3 for details). Standard Barlow Twins implementations with batch normalization exhibit similar cross-feature interactions even when $\lambda = 1$. The normalization constraints couples between different singular values, causing later-learned features to suppress earlier ones, mimicking the mixed-case dynamics. Detailed experiments are provided in Section E.4. Similarly, the behaviors of SimCLR and VICReg suggests a similar coupling dynamic. The softmax and batch normalization components likely create implicit cross-feature dependencies, leading to a phenomenon where subsequent feature learning can diminish previously learned representations. This offers a potential explanation for why both methods show sequential feature learning patterns in practice.

## 7 Practical Study

To investigate the effect of extent bias in a more realistic setting, we conducted some experiments using semi-synthetic datasets.

### 7.1 Colored-MNIST Dataset

We conducted experiments using a Colored-MNIST dataset, where we adjusted the ratio of digits pixels relative to the total image pixels. We tested three different ratios: 0.05, 0.10, and 0.15. In this dataset, we set the correlation between background and label to 70% for both training and test sets, making it difficult for a model that predicts solely based on background to achieve accuracy higher than 70%. According to our hypothesis, since backgrounds have larger extent bias than objects, the test set accuracy would rapidly increase to 70% (as the model learns background features), then plateau for a period, before slowly rising higher (as it learns object features). We also hypothesized that this plateau period would be shortened as the spatial ratio of the object increases in the images.

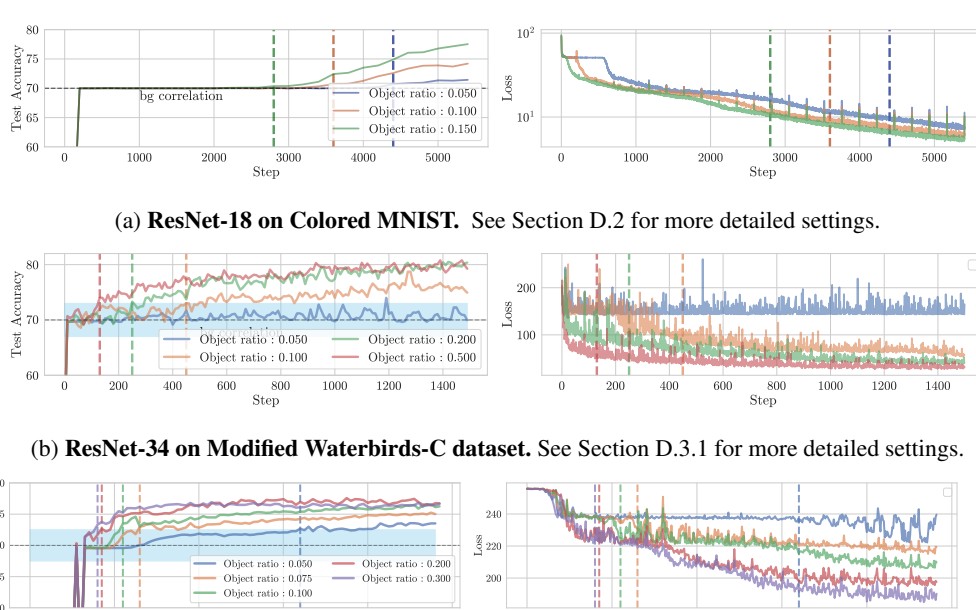

(a) **ResNet-18 on Colored MNIST.** See Section D.2 for more detailed settings.

(b) **ResNet-34 on Modified Waterbirds-C dataset.** See Section D.3.1 for more detailed settings.

(c) **ResNet-34 on Modified Waterbirds-B dataset.** See Section D.3.2 for more detailed settings.

Figure 5: **Extent bias effects on spurious dataset.** The dotted vertical lines indicate the transition points where the model shifts from background-only prediction to main object-based prediction, marked at (a) 70% accuracy with a 0.5% error tolerance (b) 70% accuracy with a $\pm3\%$ (c) 70% accuracy with a $\pm3\%$ error tolerance. (Left) The accuracy rate has a plateau at 70%, which corresponds to the correlation between background and object. The lengths of the plateaus become shorter as the object's pixel ratio increases. (Right) Loss decreases except ratio 0.05 in (b).

Figure 5a supports our hypothesis. Across all object ratio conditions (0.05, 0.10, 0.15), test accuracy exhibited a consistent pattern: a rapid increase from initial 10% to 70%, followed by a plateau period, and then a gradual ascent. Notably, as the object pixel ratio increases, the duration of the plateau phase decreases. The loss function continues to decrease even when accuracy remains stagnant at 70%. This suggests the extent bias where larger objects are prioritized during the learning process. The pattern reflects how the model initially achieves 70% accuracy by relying on background features, which statistically occupy larger regions, before progressively learning object features. Furthermore, this indicates that larger extents occupy greater eigenvalues, implying a reduction in the critical point.

## 7.2 MODIFIED WATERBIRDS DATASETS

Similarly, to test whether the extent bias observed in Colored-MNIST generalizes to more complex scenarios, we performed experiments on modified versions of Waterbirds dataset (Sagawa et al., 2019). These modified datasets comprise two distinct configurations. First, following a similar setup to Colored-MNIST, we set background colors to blue/green while varying bird sizes across multiple scales (Modified Waterbirds-C). Second, we use complex natural backgrounds (forest and sea images) while maintaining consistent bird sizes as a fixed proportion of total image pixels (Modified Waterbirds-B). We observed a similar dynamics with the Colored-MNIST experiment. See Section D.3 for details.

## 8 PRACTICAL GUIDANCE

Our analysis suggests practical strategies to mitigate shortcut learning and promote generalizable feature learning in SSL:

$\lambda$**-tuning (decorrelation strength)** In Barlow Twins, the hyperparameter $\lambda$ controls the strength of the redundancy reduction term. Our theoretical analysis across $0 \leq \lambda \leq 1$ reveals that increasing $\lambda$ facilitates the learning of smaller-extent features (with lower eigenvalues) that may be more predictive, reducing the dominance of large-extent features which may be spurious. This provides a principled approach to setting $\lambda$ beyond heuristic tuning.

**Data preprocessing design (object–background extent)** Our experiments on Colored MNIST and Modified Waterbirds demonstrate that increasing the object-to-background spatial ratio—through enlarging objects or reducing background regions—accelerates learning of predictive features and shortens shortcut-dominated phases.

**Data augmentation design (extent bias via Random Resized Crop)** Random Resized Crop (RRC) is a fundamental augmentation in SSL (Moutakanni et al., 2024). From our theoretical framework, modifying the cropping distribution directly reshapes the spectral properties of $\Gamma$, influencing which features dominate learning. We extend our extent bias framework and conduct ablation studies using synthetic datasets where features differ only in spatial extent (see Section E.5). Our experiments demonstrate that aggressive cropping amplifies learning of central features while suppressing peripheral ones, whereas milder cropping enables more balanced learning. This establishes crop aggressiveness as a key parameter for controlling extent-driven feature learning and mitigating undesired spatial biases.

## 9 CONCLUSION

In this work, we establish a theoretical connection between eigendecomposition of the feature cross-correlation matrix, shortcut learning, and stepwise learning behavior in SSL. We provide insights into how dimensional feature properties influence the learning process in SSL frameworks. This work not only explains observed shortcut learning phenomena but also offers a theoretical lens for understanding and potentially mitigating such learning biases. This theoretical framework lays the groundwork for developing more robust SSL algorithms. Future work should focus on leveraging these insights to design mechanisms that encourage learning of generalizable features despite their potentially lower extent bias or amplitude bias.

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

## A  PROOFS

### A.1  PROOF OF THEOREM 4.1

Using matrix analysis, we can express:

$$\Gamma = \mathbb{E}[x_{\text{base}}x_{\text{base}}^\top] = \begin{bmatrix} \mathbf{1}_{m_l \times m_l} & \mathbf{0}_{m_s \times m_l} \\ \mathbf{0}_{m_l \times m_s} & \mathbf{1}_{m_s \times m_s} \end{bmatrix},$$

which has two eigenvectors $e_l/\|e_l\|$ and $e_s/\|e_s\|$ corresponding to nonzero eigenvalues. We obtain the eigenvalues $m_l$ and $m_s$ from the following equation:

$$\det(\Gamma - \lambda I) = \det(\mathbf{1}_{m_l \times m_l} - \lambda I_{m_l \times m_l}) \det(\mathbf{1}_{m_s \times m_s} - \lambda I_{m_s \times m_s}) = 0.$$

Finally, we obtain the eigendecomposition $\Gamma = V_\Gamma \Lambda_\Gamma V_\Gamma$ where

$$\Lambda_\Gamma = \text{diag}\left([m_l, m_s, \ \mathbf{0}_{m-2}]\right),$$

$$V_\Gamma^{(\le d)} = \left[ \frac{1}{\sqrt{m_l}}e_l \ \ \frac{1}{\sqrt{m_s}}e_s \right].$$

### A.2  PROOF OF THEOREM 5.1

The cross-correlation matrix $\Gamma$ for this input can be expressed using (5):

$$\begin{aligned}
\Gamma &= \mathbb{E}[x_{\text{base}}x_{\text{base}}^\top] \\
&= \mathbb{E}[c_{ha}^2 b_{ha}^2 \sin(f_{ha}t)\sin(f_{ha}t)^\top + c_l^2 b_{ha}^2 \sin(f_{la}t)\sin(f_{la}t)^\top \\
&\quad + c_{ha}c_{la}b_{ha}b_{la}\sin(f_{ha}t)\sin(f_{la}t)^\top + c_{ha}c_{la}b_{ha}b_{la}\sin(f_{la}t)\sin(f_{ha}t)^\top] \\
&= c_{ha}^2 \sin(f_{ha}t)\sin(f_{ha}t)^\top + c_{la}^2 \sin(f_{la}t)\sin(f_{la}t)^\top.
\end{aligned}$$

Using the orthogonality between $\sin(f_{ha}t)$ and $\sin(f_{la}t)$ $(f_{ha} \ne f_{la})$, where $t \in \mathbb{N}$,

$$\Gamma = c_{ha}^2 \sin(f_{ha}t)\sin(f_{ha}t)^\top + c_{la}^2 \sin(f_{la}t)\sin(f_{la}t)^\top,$$

$$\Gamma \sin(f_{ha}t) = c_{ha}^2 \|\sin(f_{ha}t)\|^2 \sin(f_{ha}t),$$

$$\Gamma \sin(f_{la}t) = c_{la}^2 \|\sin(f_{la}t)\|^2 \sin(f_{la}t).$$

We find the eigenvectors and eigenvalues as:

$$\Lambda_\Gamma = \text{diag}\left([c_{ha}^2\|\sin(f_{ha}t)\|^2, c_{la}^2\|\sin(f_{la}t)\|^2, \mathbf{0}_{m-2}]\right), \ V_\Gamma^{(\le 2)} = [e_{ha} \ e_{la}]^\top.$$

With $f = \frac{2\pi}{m}k$ for some integer $k$, we have

$$\|\sin(fx)\|^2 = \int_0^m \sin^2(fx)dx = \int_0^m \frac{1-\cos(2fx)}{2}dx = \frac{1}{2}\left[x - \frac{\sin(2fx)}{2}\right]_0^m = \frac{m}{2} - \frac{\sin(2fm)}{4} = \frac{m}{2}.$$

Finally, we have

$$\Lambda_\Gamma = \text{diag}\left(\left[c_{ha}^2\frac{m}{2}, c_{la}^2\frac{m}{2}, \mathbf{0}_{m-2}\right]\right), \ V_\Gamma^{(\le 2)} = [e_{ha}/c_{ha} \ e_{la}/c_{la}].$$

### A.3  PROOF OF THEOREM 4.2

We have

$$\tilde{\lambda}_j(t) = \tilde{s}_j^2(t)\gamma_j = (1 + \lambda_j(0)^{-1}e^{-8\gamma_j t})^{-1},$$

and thus if we plug in $\tau_j = -\log(\lambda_j(0))/8\gamma_j$, i.e., $\exp(-8\gamma_j\tau_j) = \lambda_j(0)$, then we have $\tilde{\lambda}_j(\tau_j) = (1+1)^{-1} = \frac{1}{2}$. The derivative $\tilde{\lambda}_j'(t)$ at $t = \tau_j$ is given as follows:

$$\tilde{\lambda}_j'(t) = -(1 + \lambda_j(0)^{-1}e^{-8\gamma_j t})^{-2}(-8\gamma_j\lambda_j(0)^{-1}e^{-8\gamma_j t}) = -\tilde{\lambda}_j^2(t)(-8\gamma_j\lambda_j(0)^{-1}e^{-8\gamma_j t})$$

$$\tilde{\lambda}_j'(\tau_j) = -\tilde{\lambda}_j^2(\tau_j)(-8\gamma_j\lambda_j^{-1}(0)\lambda_j(0)) = 2\gamma_j.$$

Using the equations

$$C = \sum_{j=1}^{d} \lambda_j u_j u_j^\top \text{ and } C^2 = \sum_{j=1}^{d} \lambda_j^2 u_j u_j^\top,$$

we get the loss

$$\mathcal{L} = \|C - I\|_F^2 = \text{Tr}((C - I)(C - I)) = \text{Tr}(C^2) - 2\,\text{Tr}(C) + d$$

$$= \sum_{j=1}^{d} \lambda_j^2 - 2\sum_{j=1}^{d} \lambda_j + d = \sum_{j=1}^{d} (\lambda_j - 1)^2.$$

Thus, we get the following equation:

$$\frac{d\mathcal{L}}{dt}(\tau_j) = \sum_{i=1}^{d} 2(\lambda_i(\tau_j) - 1)\lambda_i'(\tau_j)$$

$$\approx \sum_{i=1}^{d} 2(\tilde{\lambda}_i(\tau_j) - 1)\tilde{\lambda}_i'(\tau_j)$$

$$\approx 2(\tilde{\lambda}_j(\tau_j) - 1)\tilde{\lambda}_j'(\tau_j)$$

$$= -\tilde{\lambda}_j'(\tau_j) = -2\gamma_j.$$

## A.4    PROOF OF THEOREM 4.3

First, we have

$$\tilde{s}_j(t) = (\gamma_j + s_j^{-2}(0)\exp(-8\gamma_j t))^{-1/2},$$

$$\tilde{s}_j(\tau_j) = (\gamma_j + s_j^{-2}(0)\lambda_j(0))^{-1/2} = (2\gamma_j)^{-1/2}.$$

and its derivative is given as follows:

$$\tilde{s}_j'(t) = -\frac{1}{2}(\gamma_j + s_j^{-2}(0)\exp(-8\gamma_j t))^{-3/2}(-8\gamma_j s_j^{-2}(0)\exp(-8\gamma_j t)),$$

$$\tilde{s}_j'(\tau_j) = -\frac{1}{2}(\gamma_j + s_j^{-2}(0)\lambda_j(0))^{-3/2}(-8\gamma_j s_j^{-2}(0)\lambda_j(0))$$

$$= -\frac{1}{2}(2\gamma_j)^{-3/2}(-8\gamma_j^2)$$

$$= (2\gamma_j)^{1/2}.$$

## A.5    PROOF OF DEEP LINEAR LAYER

In Deep Linear Networks (DLNs), We assume that, $\tilde{W} = W_L W_{L-1} \cdots W_2 W_1, \forall W_k \in \mathbb{R}^{m \times m}$. Under the toy Barlow Twins loss $\mathcal{L} = \|\tilde{W}\Gamma\tilde{W}^\top - I_m\|_F^2$, each layer has gradient of:

$$\frac{\partial \mathcal{L}}{\partial W_k} = -4\left(\prod_{j=k+1}^{L} W_j\right)^\top (\tilde{W}\Gamma\tilde{W}^\top - I_d)\tilde{W}\Gamma\left(\prod_{j=1}^{k-1} W_j\right)^\top$$

If we assume: $W_k W_k^\top = W_{k+1}^\top W_{k+1}, \ \forall k \in [1, L-1]$ we derive same singular value on every layer,

$$W_k = U_k S U_{k+1}^\top, \quad \forall k \in [1, L-1]$$

where $S = \frac{1}{\sqrt{L}}\text{diag}(\sigma_1, \sigma_2, \ldots, \sigma_d)$, $U_1 = \tilde{U}$, $U_{L+1} = \tilde{V}$. We assume the total weight $\tilde{W} = \tilde{U}\tilde{S}\tilde{V}^T$, $V = V_\Gamma$

$$\prod_{j=k+1}^{L} W_j = \prod_{j=k+1}^{L} U_j S U_{j+1}^\top = U_{k+1} S^{L-k} \tilde{V}^\top,$$

$$\prod_{j=1}^{k-1} W_j = \prod_{j=1}^{k-1} U_j S U_{j+1}^\top = \tilde{U} S^{k-1} U_k^\top.$$

$$\tilde{W}\Gamma = \tilde{U}S^L \tilde{V}^\top \Gamma = \tilde{U}S^L \tilde{V}^T V_\Gamma \Lambda_\Gamma V_\Gamma^\top = \tilde{U}S^L \Lambda_\Gamma V_\Gamma^\top$$

$$\tilde{W}\Gamma\tilde{W}^T = \tilde{U}S^L \Lambda_\Gamma S^L \tilde{U}^T = \tilde{U}S^{2L}\Lambda_\Gamma \tilde{U}^T$$

$$\frac{\partial \mathcal{L}}{\partial W_k} = -4(U_{k+1}S^{L-k}\tilde{V}^\top)^\top(\tilde{U}S^{2L}\Lambda_\Gamma \tilde{U}^\top - I_d)(\tilde{U}S^L \Lambda_\Gamma V_\Gamma^\top)(\tilde{U}S^{k-1}U_k^\top)^\top$$

$$= -4(\tilde{V}S^{L-k}U_{k+1}^\top)(\tilde{U}S^{2L}\Lambda_\Gamma \tilde{U}^\top - \tilde{U}I_d\tilde{U}^\top)(\tilde{U}S^L \Lambda_\Gamma V_\Gamma^\top)(U_k S^{k-1}\tilde{U}^\top)$$

$$= -4(\tilde{V}S^{L-k}U_{k+1}^\top)\tilde{U}(S^{2L}\Lambda_\Gamma - I_d)\tilde{U}^\top(\tilde{U}S^L \Lambda_\Gamma V_\Gamma^\top)(U_k S^{k-1}\tilde{U}^\top)$$

$$= -4(\tilde{V}S^{L-k}U_{k+1}^\top)\tilde{U}(S^{2L}\Lambda_\Gamma - I_d)S^L \Lambda_\Gamma V_\Gamma^\top(U_k S^{k-1}\tilde{U}^\top)$$

For analytical tractability, we assume $W_k(0) = S_0 I, \tilde{W}_k(0) = S_0^L I = \tilde{S}_0 I$. Under this condition, $U_k(0)V_k(0)^\top = I, \tilde{U}(0)\tilde{V}(0)^\top = I$.

$$\frac{\partial \mathcal{L}}{\partial W_k} = -4(\tilde{V}S^{L-k}U_{k+1}^\top)\tilde{U}(S^{2L}\Lambda_\Gamma - I_d)S^L \Lambda_\Gamma \tilde{V}^\top(U_k S^{k-1}\tilde{U}^\top)$$

$$= -4\tilde{V}S^{L-k}(S^{2L}\Lambda_\Gamma - I_d)S^L \Lambda_\Gamma S^{k-1}\tilde{U}^\top$$

$$= -4\tilde{V}(S^{2L}\Lambda_\Gamma - I_d)S^{2L-1}\Lambda_\Gamma \tilde{U}^\top$$

$$= -4\sum_i (\sigma_i^{2L}\gamma_i - 1)\sigma_i^{2L-1}\gamma_i v_i u_i^\top$$

Using chain rule,

$$\frac{\partial \mathcal{L}}{\partial \sigma_j} = \mathrm{Tr}\left[\left(\frac{\partial \mathcal{L}}{\partial W_k}\right)^\top \frac{\partial W_k}{\partial \sigma_j}\right] = \mathrm{Tr}\left[\left(-4\sum_i (\sigma_i^{2L}\gamma_i - 1)\sigma_i^{2L-1}\gamma_i v_i u_i^\top\right)u_j v_j^\top\right]$$

$$= \mathrm{Tr}\left[-4(\sigma_i^{2L}\gamma_i - 1)\sigma_i^{2L-1}\gamma_i\right] = -4(\sigma_i^{2L}\gamma_i - 1)\sigma_i^{2L-1}\gamma_i$$

$$\tilde{\sigma}_j = (\sigma_j)^L$$

$$\frac{d\tilde{\sigma}_j}{dt} = L(\sigma_j)^{L-1}\frac{d\sigma_j}{dt} = -L(\sigma_j)^{L-1}\frac{\partial \mathcal{L}}{\partial \sigma_j}$$

$$= L(\sigma_j)^{L-1}4\sigma_j^{2L-1}\gamma_j(1 - \sigma_j^{2L}\gamma_j) = 4L\sigma_j^{3L-2}\gamma_j(1 - \sigma_j^{2L}\gamma_j) = 4L\tilde{\sigma}_j^{(3L-2)/L}\gamma_j(1 - \tilde{\sigma}_j^2\gamma_j)$$

$$= 4L\tilde{\sigma}_j^{3-2/L}\gamma_j(1 - \tilde{\sigma}_j^2\gamma_j) \tag{9}$$

### A.6 Proof of $\lambda$ effect in Barlow twins

First we consider general Barlow Twins loss:

$$L_\lambda = \sum_i ([W\Gamma W^\top]_{ii} - 1)^2 + \lambda \sum_{i\neq j}[W\Gamma W^\top]_{ij}^2 = (1-\lambda)L_0 + \lambda L_1.$$

Thus it exhibits a mixed dynamics between the $L_0$ and $L_1$. Therefore, we first consider the dynamics of $L_0$ and $L_1$:

$L_1$ **case**

$$L_1 = ||W\Gamma W^\top - I_d||_F^2 = \sum_j (s_j^2\gamma_j - 1)^2,$$

$$\frac{dL_1}{ds_k} = 4s_k\gamma_k(\sum_j \delta_{kj}s_j^2\gamma_j - 1).$$

$L_0$ **case**

$$L_0 = \sum_i ([W\Gamma W^\top]_{ii} - 1)^2 = \sum_i (e^{(i)\top} W\Gamma W^\top e^{(i)} - 1)^2 = \sum_i (u^{(i)\top} S\Lambda_\Gamma S^\top u^{(i)} - 1)^2$$

$$= \sum_i (\sum_j u_j^{(i)2} s_j^2 \gamma_j - 1)^2$$

$$\frac{dL_0}{ds_k} = \sum_i 2(\sum_j u_j^{(i)2} s_j^2 \gamma_j - 1) u_k^{(i)2} 2 s_k \gamma_k = 4 s_k \gamma_k \sum_i (\sum_j u_j^{(i)2} s_j^2 \gamma_j - 1) u_k^{(i)2}.$$

If $u_j^{(i)2} = \delta_{ij}$ ($U = I_d$), then $L_0 = L_1$. If not, $\sum_j u_j^{(i)2} = 1$ and each $u_j^{(i)2}$ acts as an averaging weight $1/d$.

We now investigate the dynamics of $s_k$'s: Initially, all singular values grow with exponential dynamics as $s_j \approx 0$ and $\delta_{kj} = 0$ for $j \neq k$.

$$\dot{s}_k = -\frac{dL_0}{ds_k} = -4 s_k \gamma_k \sum_i (\sum_j u_j^{(i)2} s_j^2 \gamma_j - 1) u_k^{(i)2} = 4 s_k \gamma_k + O(s_k^3)$$

$$\dot{s}_k = -\frac{dL_1}{ds_k} = -4 s_k \gamma_k (\sum_j \delta_{kj} s_j^2 \gamma_j - 1) = 4 s_k \gamma_k + O(s_k^3)$$

After a few steps, the first singular value $s_1$ increases (since $\gamma_1$ is the largest) and then,

$$\dot{s}_1 = -\frac{dL_0}{ds_1} = -4 s_1 \gamma_1 \sum_i (\sum_j u_j^{(i)2} s_j^2 \gamma_j - 1) u_1^{(i)2} = 4 s_1 \gamma_1 \left( 1 - \sum_j (\sum_i u_j^{(i)2} u_1^{(i)2}) s_j^2 \gamma_j \right)$$

$$= 4 s_1 \gamma_1 \left( 1 - \sum_i u_1^{(i)4} s_1^2 \gamma_1 \right) + O\left( \max_{j>1} s_j^2 \right) \approx 4 s_1 \gamma_1 \left( 1 - \frac{1}{d} s_1^2 \gamma_1 \right)$$

$$\dot{s}_1 = -\frac{dL_1}{ds_1} = -4 s_1 \gamma_1 (\sum_j \delta_{1j} s_j^2 \gamma_j - 1) = 4 s_1 \gamma_1 \left( 1 - s_1^2 \gamma_1 \right)$$

[$\lambda = 0$] $s_1$ saturates as $s_1^2 \gamma_1$ reaches $d$.

[$0 < \lambda < 1$] $s_1$ saturates as $s_1^2 \gamma_1$ reaches the harmonic mean $\frac{1}{\lambda + (1-\lambda)\frac{1}{d}}$ of 1 and $d$.

[$\lambda = 1$] $s_1$ saturates as $s_1^2 \gamma_1$ reaches 1.

After the first loss drops where $s_k$'s are still small except for $s_1$

$$\dot{s}_k = -\frac{dL_0}{ds_k} = -4 s_k \gamma_k \sum_i (\sum_j u_j^{(i)2} s_j^2 \gamma_j - 1) u_k^{(i)2} = O(s_k)$$

$$\dot{s}_k = -\frac{dL_1}{ds_k} = -4 s_k \gamma_k (\sum_j \delta_{kj} s_j^2 \gamma_j - 1) = 4 s_k \gamma_k (1 - s_k^2 \gamma_k)$$

[$\lambda = 0$] $\dot{s}_k$ becomes nearly zero, effectively stopping the growth of other singular values. Each $s_k$ ($k \neq 1$) stays near zero.

[$0 < \lambda < 1$] First, $s_k$ exponentially grows with the following dynamics:

$$\dot{s}_k = -4 s_k \gamma_k \sum_i (\sum_j u_j^{(i)2} s_j^2 \gamma_j - 1) u_k^{(i)2} = 4 s_k \gamma_k \left( 1 - \sum_i \sum_{j \neq k} u_j^{(i)2} u_k^{(i)2} s_j^2 \gamma_j \right) + O(s_k^3)$$

with the growth rate smaller than that of $\lambda = 1$:

$$4 \gamma_k \left( 1 - \sum_i \sum_{j \neq k} u_j^{(i)2} u_k^{(i)2} s_j^2 \gamma_j \right) < 4 \gamma_k.$$

Then, each $s_k$ follows sigmoidal dynamics and saturates near some value larger than $\sqrt{1/\gamma_k}$. But after saturation $s_k$ decreases again since the dynamics of $s_k$ is coupled with other singular values. As other singular value increases, the sum $\sum_i \sum_{j \neq k} u_j^{(i)2} u_k^{(i)2} s_j^2 \gamma_j$ grows and exceeds 1, which causes $s_k$ to decrease as the exponent becomes negative. Note that the principle "features with larger eigenvalues are learned earlier" still holds.

[$\lambda = 1$] Each $s_k$ exhibits independent sigmoidal dynamics and saturates as $s_k^2 \gamma_k$ reaches 1.

$\lambda = 0$ **Case** Dynamics of $s_k(t)$ is almost only $s_1$ changes. In the beginning $s_1$ shows sigmoidal dynamics, increases and saturates near $\sqrt{d/\gamma_1}$. $\tau_1 \sim \frac{-\log(s_1^2(0)\gamma_1/d)}{8\gamma_1}$. After the first drop $\dot{s}_k \approx 0$ and $s_k$ stays near zero for $k = 2, 3, \cdots$. $\tau_k = \infty$

$\lambda = 1$ **Case** Dynamics of $s_k(t)$ is each $s_k$ affects each other. In the beginning $s_1$ shows sigmoidal dynamics, increases and saturates at $\sqrt{\frac{1}{(\lambda + (1-\lambda)\frac{1}{d})\gamma_1}}$. $\tau_1 \sim \frac{-\log(s_1^2(0)\gamma_1)}{8\gamma_1}$. After the first drop $s_k$ shows independent sigmoidal dynamics, increases and saturates at $\sqrt{1/\gamma_k}$, starting from smaller $k = 2, 3, \cdots$. $\tau_k \propto \frac{1}{\gamma_k}$

$0 < \lambda < 1$ **Case** In case of loss $= (1 - \lambda)L_0 + \lambda L_1$, dynamics of $s_k(t)$ is independent sigmoidal dynamics. In the beginning $s_1$ shows sigmoidal dynamics, increases and saturates at $\sqrt{1/\gamma_1}$. $\tau_1 \sim \frac{-\log\left(s_1^2(0)\gamma_1(\lambda + (1-\lambda)\frac{1}{d})\right)}{8\gamma_1}$. After the first drop $s_k$ increases sigmoidally and saturates near some value larger than $\sqrt{1/\gamma_k}$ (and decreases slowly when the next singular value $s_{k+1}$ increases) $\tau_k$ relatively later than the $\lambda = 1$ case.

# B  MAIN RESULTS EXPERIMENT

## B.1  LINEAR NETWORK

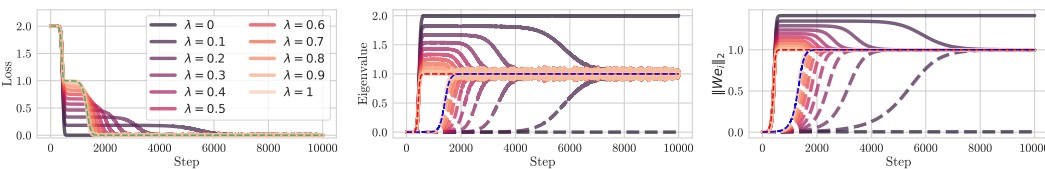

Figure 6: **Effect of redundancy reduction coefficient $\lambda$ on linear network.** Experimental investigation of extent bias.

**Extent bias**    We train a linear layer with a batch size of 1000. $m_l = 6, m_s = 2$. Learning rate $\eta = 8 \times 10^{-4}$, a scaling factor is $5 \times 10^{-4}$. We trained 10,000 steps.

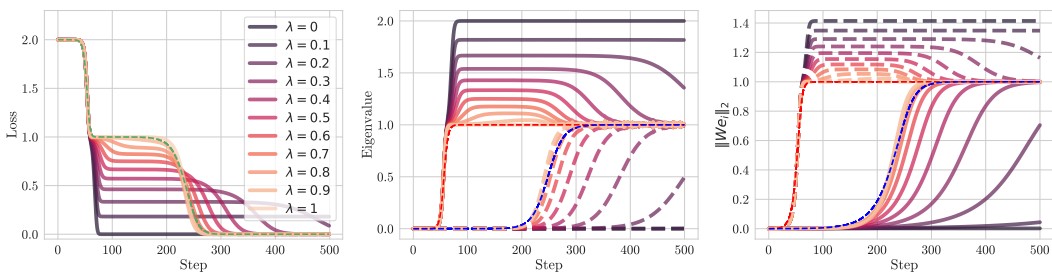

Figure 7: **Effect of redundancy reduction coefficient $\lambda$ on linear network.** Experimental investigation of amplitude bias.

**Amplitude bias**    We trained a 4-layer mlp with leaky ReLU as activation function. We use a batch size of 1000. The hidden layer width is the same as the input size. $m_l = 6, m_s = 2$. Learning rate $\eta = 10^{-3}$, a scaling factor is $10^{-4}$. We trained 500 steps.

### B.1.1 DETAILED THEORETICAL RESULT ON EXTENT BIAS

**Orthogonal Feature Learning** Our analysis shows that features are learned as orthogonal to each other, where each feature is acquired independently without interference from others. This orthogonal learning pattern is particularly evident in the evolution of the model's weight matrix singular vectors. To formalize this observation, we analyze how the left singular vectors of the weight matrix align with the feature vectors during training.

**Theorem B.1.** *Under Assumption 1, the left singular vectors $u$ of $W(t)$ learn features orthogonally:*

$$Proj_{U(\leq 2)}(We_l) := (u_l^\top We_l, u_s^\top We_l) = (\sqrt{\lambda_l}, 0),$$

$$Proj_{U(\leq 2)}(We_s) := (u_l^\top We_s, u_s^\top We_s) = (0, \sqrt{\lambda_s}),$$

*where $u_l, u_s$ are the corresponding left singular vectors for the singular values $s_l, s_s$.*

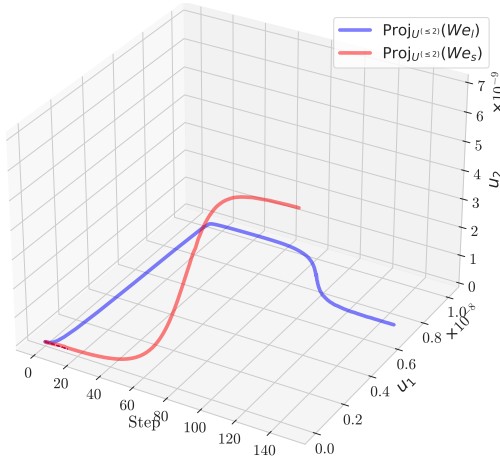

Figure 8: **Visualization of the trajectory of $We_l$ and $We_s$ on the subspace spanned by $u_1, u_2$ during training.** The high-dimensional feature $We_h$ (blue solid line) aligns with $u_1$ and the low-dimensional feature $We_l$ (red solid line) aligns with $u_2$. Dashed lines are predicted trajectory (see Theorem B.1).

Figure 8 shows orthogonal learning pattern that features are learned independently and sequentially, supporting our theoretical analysis of stepwise learning dynamics.

**Aligned Initialization and Subspace Alignment**    While Assumption 1 assumes perfect alignment at initialization, Simon et al. (2023) demonstrate that this assumption can be relaxed significantly. They show that even with generic small random initialization, the dynamics quickly converge to the aligned trajectory. This result significantly strengthens our analysis by showing that the aligned initialization assumption is not restrictive, any sufficiently small initialization will rapidly align with the top eigenvectors of $\Gamma$ before substantial feature learning begins.

To validate this theoretical assumption, we measure the subspace alignment metric as:

**Definition 1** (Subspace Alignment). We define subspace alignment of $\text{Im}(A)$ and $\text{Im}(B)$:

$$\text{SA}(A, B) = ||A^\top B||_F^2/d,$$

where $\text{Im}(A) = \{Av \in \mathbb{R}^m : v \in \mathbb{R}^d\}$, $A = [a_1 \cdots a_d], B = [b_1 \cdots b_d] \in \mathbb{R}^{m \times d}$, $a_i \perp a_j$, $b_i \perp b_j$ $(i \neq j)$ and $a_i, b_i \in \mathbb{R}^m$ are unit vectors.

Note that $0 \leq \text{SA}(A, B) \leq 1$ and it attains $\text{SA}(A, B) = 0$ when $\text{Im}(A) \perp \text{Im}(B)$, and $\text{SA}(A, B) = 1$ when $\text{Im}(A) = \text{Im}(B)$. Figure 14 (Top) in Section B.1.3 empirically validates Assumption 1 using the subspace alignment metric. The model becomes aligned rapidly in the early stages of training, satisfying the assumption.

### B.1.2 DETAILED THEORETICAL RESULT ON AMPLITUDE BIAS

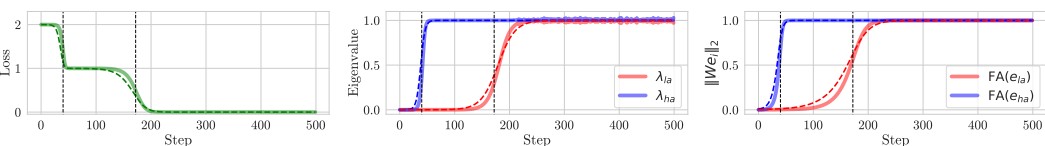

Figure 9: **Amplitude bias effects on learning dynamics in SSL.** See the caption of Figure 1. Note that the time steps $t_1$ and $t_2$ with $t_1 : t_2 \approx \frac{1}{\gamma_h} : \frac{1}{\gamma_l} = \frac{1}{c_h^2} : \frac{1}{c_l^2}$. We use $c_h = 1, \ c_l = 1/2$. See Section B.1.2 for more detailed settings.

**Amplitude Experiment**    For the amplitude experiment shown in Section 5.1, we train the model using 500 steps. The augmentation noise parameter $a$ is set to $0.1$. We use a dataset size of $n = 1000$ samples with feature frequency $f_{ha} = 2\frac{2\pi}{24}, f_{la} = 32\frac{2\pi}{24}$. We also use a learning rate $\eta = 5 \cdot 10^{-5}$, a scaling factor $3 \cdot 10^{-3}$ and $m = 96$.

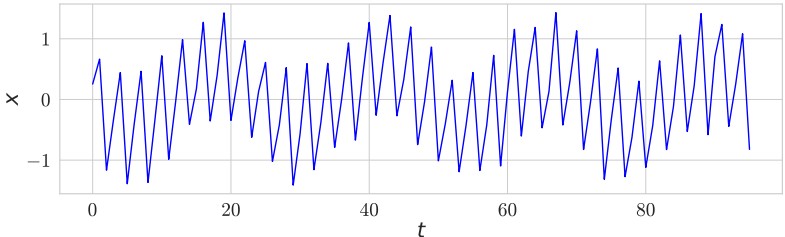

Figure 10: **Input data** $x = x_{base} + \epsilon$. $x_{\text{base}}[t] = b_{ha}c_{ha}\sin(f_{ha}t) + b_{la}c_{la}\sin(f_{la}t)$, where $c_{ha} = 1, c_{la} = 0.5, f_{ha} = \frac{2\pi}{m}32, f_{la} = \frac{2\pi}{m}8, m = 96$.

**Right Singular Vectors of** $W$

**Cross-Correlation eigenvalue** $\lambda$ **and Loss Relationship**    We analyze how the eigenvalues $\lambda$ relate to the loss dynamics. The relationship follows similar patterns to those observed in Section 4.3, but with coefficients $c_h$ and $c_l$ rather than $m_l$ and $m_s$.

Figure 17 in Section B.1.4 shows the close relationship between the derivatives of cross-correlation eigenvalues $\frac{d\lambda_{ha}}{dt}$, $\frac{d\lambda_{la}}{dt}$ and $\frac{d\mathcal{L}}{dt}$. The peaks in these derivatives occur at the critical points with

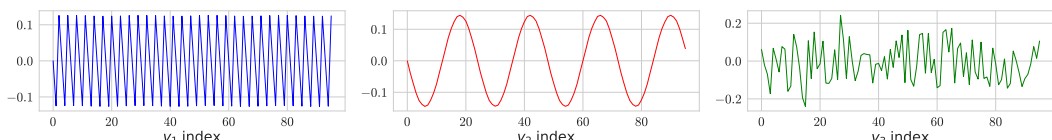

Figure 11: **The eigenvectors $v_i$'s of $\Gamma$ for $i = 1, 2, 3$ (from Left to Right).** (Left) The first eigenvector that corresponds to the largest eigenvalue indicates the (high frequency) feature with a high amplitude $c_{ha} \sin(f_{ha}t)$, (Center) the second the (low frequency) feature with a low amplitude feature $c_{la} \sin(f_{la}t)$, (Right) the third (and beyond) noise, where $c_{la} < c_{ha}$.

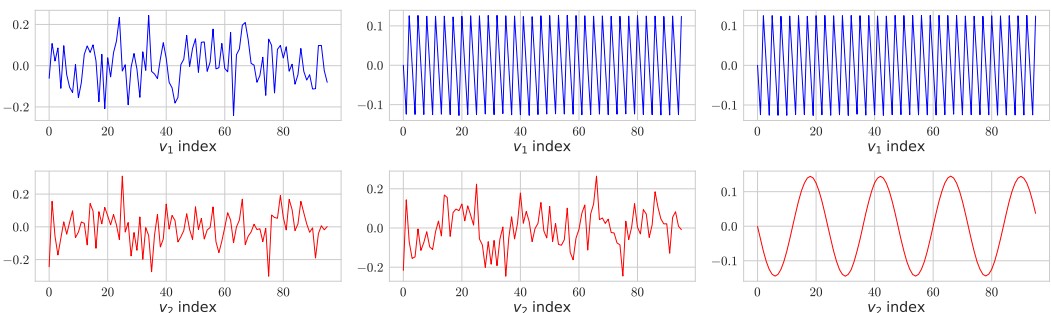

Figure 12: **The first two right singular vectors (Top/Bottom) of $W$ during training (from Left to Right).** (Left) At $t = 0$, the two singular vectors are just noise. (Center) A little after $t = \tau_1$, the first singular value reaches the plateau as shown in Figure 9 and only the (high frequency) feature with a high amplitude is learned. (Right) At the convergence, the model learns the two features.

magnitudes proportional to the corresponding coefficients $\gamma_{ha} : \gamma_{la} = c_{ha}^2 : c_{la}^2$ (see (8)). This shows our theoretical predictions Theorem 4.2 matches empirical result.

**Weight Singular Value Evolution**   We analyze how the singular values of the weight matrix evolve during training. Similarly to the extent bias case, we expect the singular values $s_j$ to converge to theoretical limits determined by the feature coefficients.

Figure 18 in Section B.1.4 shows the evolution of singular values $s_{ha}$ and $s_{la}$ of weight matrix $W$ (Left) and their derivatives (Right). The singular values converge to their theoretical limits $1/\sqrt{\gamma_j}$ predicted by Theorem 4.3, where $\gamma_j = c_j^2 \frac{m}{2}$. At the critical points $\tau_j$, the derivatives achieve their maximum values of $\sqrt{2\gamma_j}$, showing that rates of feature learning are proportional to the coefficients. These results confirm that the feature coefficients, rather than their frequencies, govern both the convergence values and rates of feature learning.

**Aligned Initialization and Subspace Alignment**   To validate Assumption 1 about alignment between the weight matrix singular vectors and eigenvectors of $\Gamma$, we measure the subspace alignment metric as defined in the extent case Definition 1. Figure 14 (Bottom) in Section B.1.3 empirically validates our assumption through subspace alignment measurements. As discussed in Section B.1.1, the model achieves alignment rapidly in the early stages of training, even with small random initializations.

**Orthogonal Feature Learning**   Similar to the extent case, we investigate how the weight matrix learns different frequency components orthogonally as shown in Theorem B.1. The orthogonal learning pattern reveals how frequency features are acquired independently despite their different spectral characteristics.

Figure 13 shows the trajectories of weight matrix in terms of their alignments with frequency components $e_{ha}$ and $e_{la}$. The blue trajectory shows the first learning phase where $u_1$ aligns with the high-amplitude feature ($c_{ha} \sin(f_{ha}t)$), followed by the red trajectory showing $u_2$ aligning with the low-amplitude feature ($c_{la} \sin(f_{la}t)$). This sequential, orthogonal learning pattern demonstrates that

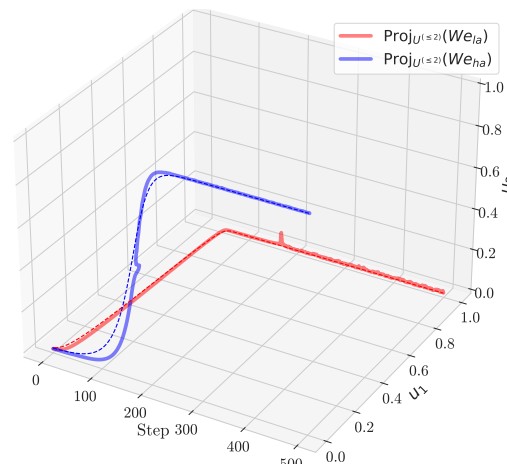

Figure 13: **Visualization of the trajectory of $We_{ha}$ and $We_{la}$ on the subspace spanned by $u_1, u_2$ during training.** See the caption of Figure 8.

feature learning is primarily determined by coefficient magnitudes rather than frequency characteristics, supporting our analysis in Theorem B.1.

### B.1.3 SUBSPACE ALIGNMENT

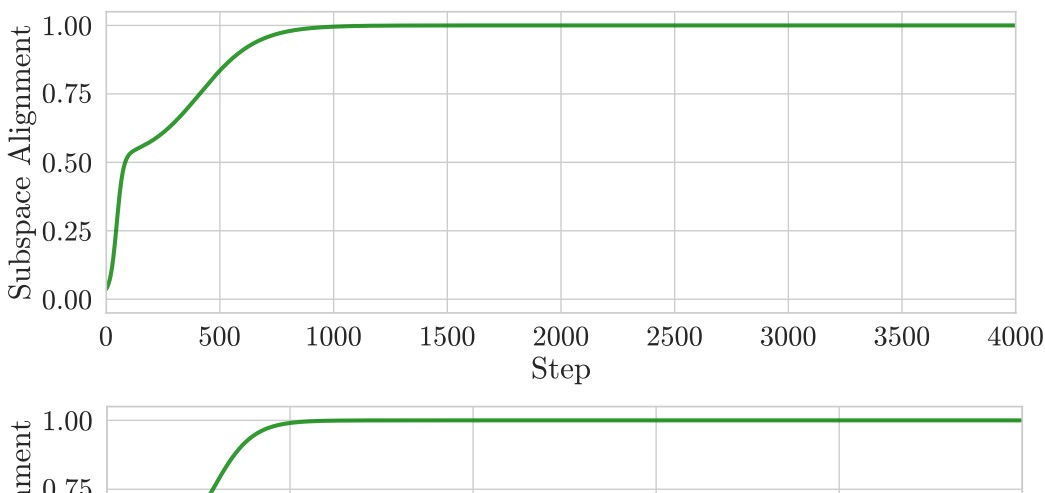

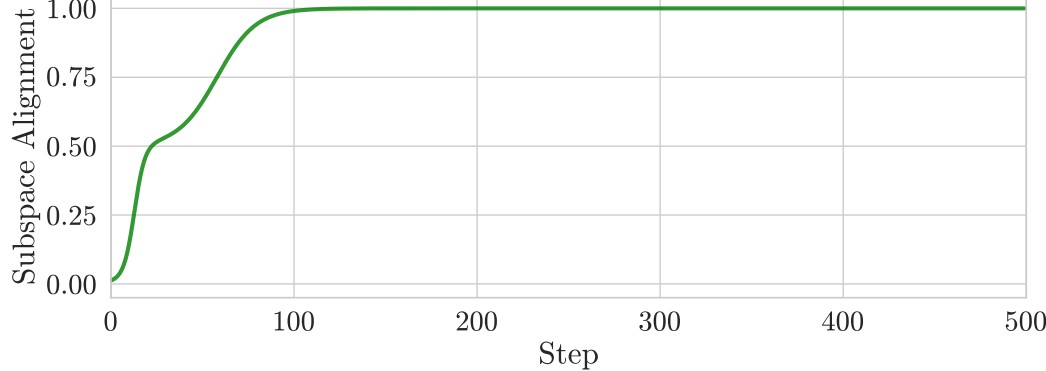

Figure 14: Evolution of subspace alignment $\text{SA}(V^{(\leq d)}, V_\Gamma^{(\leq d)})$ ($d = 2$) between the top-$d$ right singular vectors of $W$ and eigenvectors of $\Gamma$. We use data (Top) from Section 4.1 and (Bottom) from Section 5.1. See Section D.

### B.1.4 DERIVATIVES

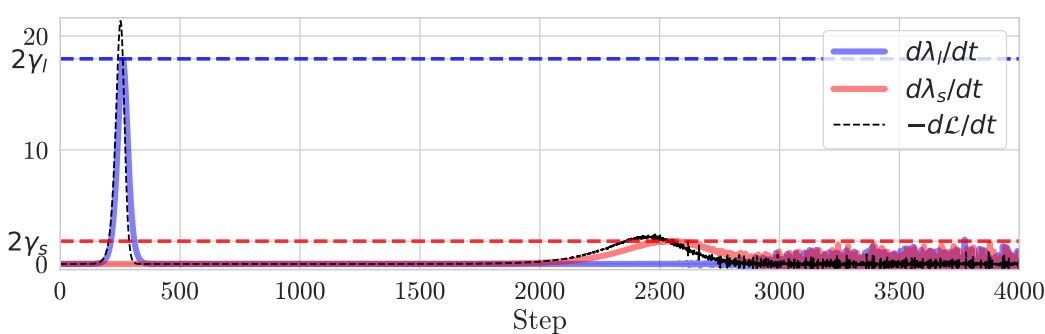

Figure 15: **Derivatives** $\frac{d\lambda_l}{dt}$ **(blue)**, $\frac{d\lambda_s}{dt}$ **(red), and** $-\frac{d\mathcal{L}}{dt}$ **(black dashed).** The derivative $\frac{d\lambda_l}{dt}(\tau_l)$ (solid blue), $\frac{d\lambda_s}{dt}(\tau_s)$ (solid red) are approximately equal to $2\gamma_l = 2m_l$ (dashed blue), $2\gamma_s = 2m_s$ (dashed red).

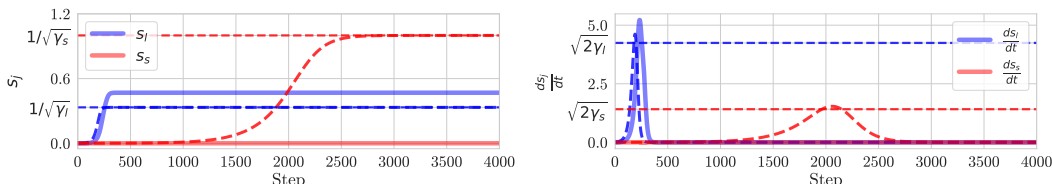

Figure 16: **Evolution of** $s_j(t)$ **and** $s'_j(t)$**.** (Left) Evolution of singular values $s_l$ (solid blue) and $s_s$ (solid red) of $W$ during training. They converge near to $1/\sqrt{\gamma_l} = 1/3$ (dashed horizontal blue) and $1/\sqrt{\gamma_s} = 1$ (dashed horizontal red), respectively. The predicted singular values (dashed blue, dashed red) match the empirical result. (Right) Evolution of the derivatives $\frac{ds_l}{dt}$ (solid blue) and $\frac{ds_s}{dt}$ (solid red). The derivatives $\frac{ds_l}{dt}(\tau_l)$, $\frac{ds_s}{dt}(\tau_s)$ are approximately equal to $\sqrt{2\gamma_l}$ (dashed horizontal blue), $\sqrt{2\gamma_s}$ (dashed horizontal red). The predicted derivatives of singular values (dashed blue, dashed red) also match the empirical result. We use $m_l = 9$ and $m_s = 1$.

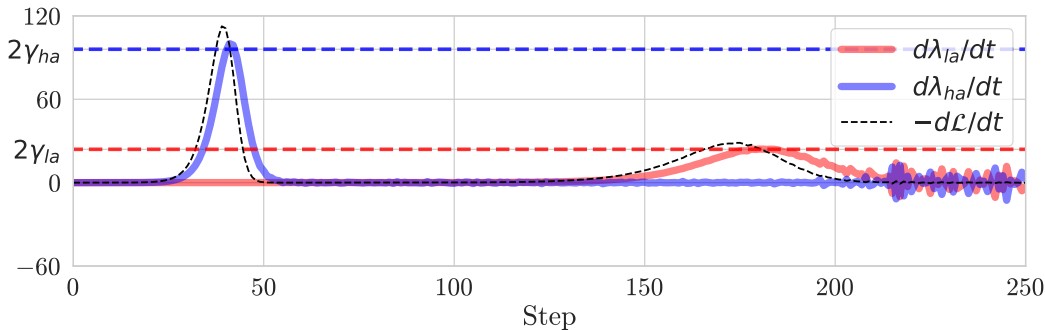

Figure 17: **Derivatives** $\frac{d\lambda_{ha}}{dt}$ **(blue),** $\frac{d\lambda_{la}}{dt}$ **(red), and** $-\frac{d\mathcal{L}}{dt}$ **(black dashed).** The derivative $\frac{d\lambda_{ha}}{dt}(\tau_{ha})$ (solid blue), $\frac{d\lambda_{la}}{dt}(\tau_{la})$ (solid red) are approximately equal to $2\gamma_{ha} = 2c_{ha}^2$(dashed blue), $2\gamma_{la} = 2c_{la}^2$(dashed red). See Figure 15 together.

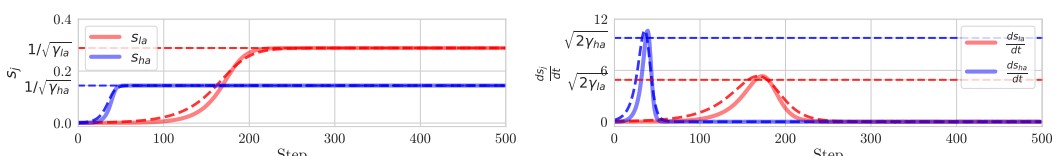

Figure 18: **Evolution of** $s_j(t)$ **and** $s_j'(t)$**.** See the caption of Figure 16. (Left) They converge near to $1/\sqrt{\gamma_{ha}} = 1/\sqrt{c_{ha}^2 \frac{m}{2}}$ and $1/\sqrt{\gamma_{la}} = 1/\sqrt{c_{la}^2 \frac{m}{2}}$ . (Right) The derivatives $\frac{ds_{ha}}{dt}(\tau_{ha})$, $\frac{ds_{la}}{dt}(\tau_{la})$ are approximately equal to $\sqrt{2\gamma_{ha}}$, $\sqrt{2\gamma_{la}}$. We use $c_{ha} = 1$ and $c_{la} = 1/2$.

## B.2 DEEP LINEAR NETWORK

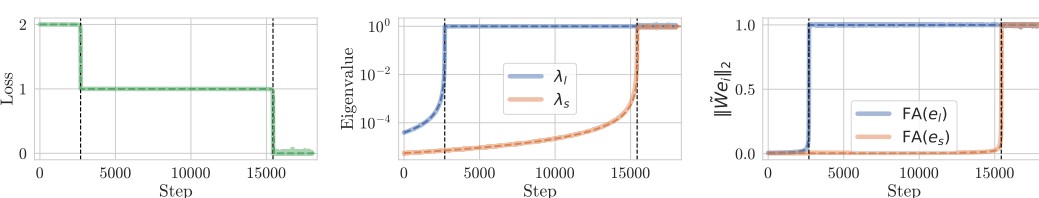

Figure 19: Ideal initialization condition in DLN, assumed in Section A.5. (Left) Stepwise learning curves of Barlow Twins. The predicted loss (dashed green) of $\mathcal{L} = \sum_{j=1}^{d}(\tilde{\lambda}_j(t) - 1)^2 = \sum_{j=1}^{d}(\tilde{s}_j^2(t)\gamma_j - 1)^2$ using (3) match the empirical result (solid green) (Center) Evolution of eigenvalues $\lambda_j$'s of $C$ during training. We compare them with the predicted evolution $\tilde{\lambda}_j(t)$ (dashed lines). (Right) Evolution of the feature alignment $||We||_2$ for $e = e_l$ (blue) and $e = e_s$ (red). It shows very similar behaviors to the eigenvalues $\tilde{\lambda}_j^{1/2}$ (dashed lines).

**Ideal Deep Linear Network**  Under ideal initialization conditions assumed in Section A.5, established to verify (9), the results are in exact agreement with the predictions in Figure 19.

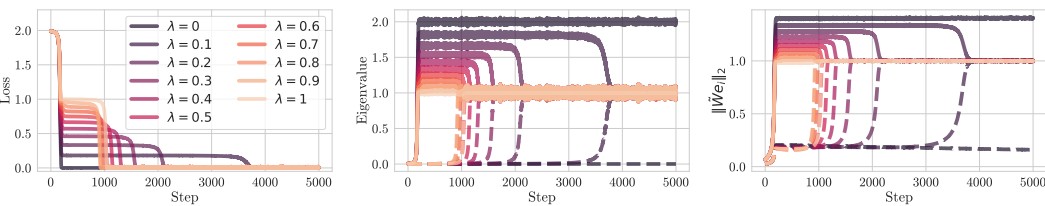

Figure 20: Training loss and eigenvalue of DLN on extent bias. (Left) Training loss curve showing the relationship between $\lambda$ and convergence behavior. (Right) Corresponding eigenvalue variations demonstrating the impact of the redundancy reduction term.

**Extent bias**  We trained 4 linear layers. We use a batch size of 1000, The hidden layer width is the same as the input size. $m_l = 6, m_s = 2$. Learning rate $\eta = 10^{-2}$, a scaling factor is $6 \times 10^{-1}$. We trained 18,000 steps.

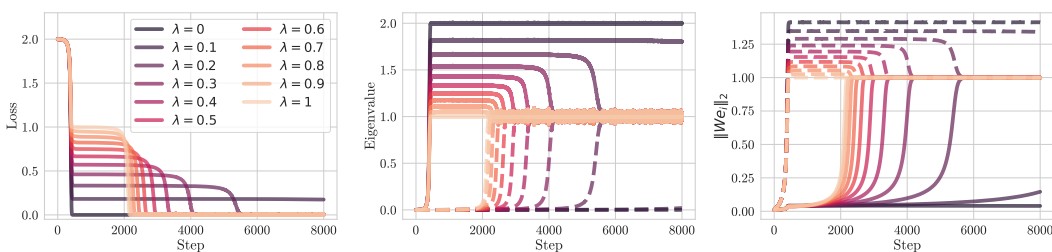

Figure 21: Training loss and eigenvalue of DLN on amplitude bias. (Left) Training loss curve showing the relationship between $\lambda$ and convergence behavior. (Right) Corresponding eigenvalue variations demonstrating the impact of the redundancy reduction term.

**Amplitude bias**  We trained 4 linear layers. We use batch size as 2000, The hidden layer width is 8. $m_l = 6, m_s = 2$. Learning rate $\eta = 1 \times 10^{-3}$, a scaling factor is $4 \times 10^{-1}$. We trained 8,000 steps.

### B.3 MULTI-LAYER PERCEPTRON

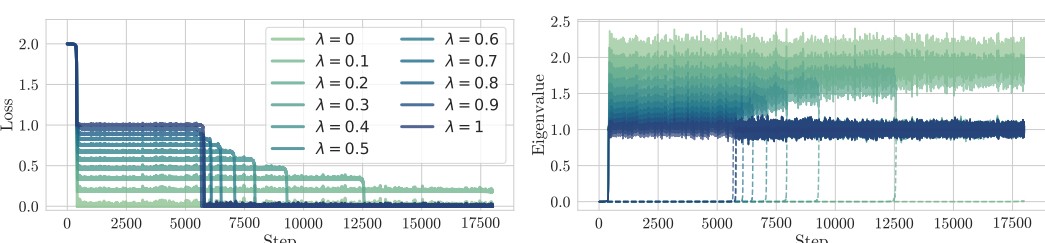

Figure 22: Training loss and eigenvalues of an MLP network with Leaky ReLu as activation on extent bias. (Left) Training loss curve showing the relationship between $\lambda$ and convergence behavior. (Right) Corresponding eigenvalue variations demonstrating the impact of the redundancy reduction term on.

**Extent bias**   We trained a 4-layer mlp with leaky ReLU as activation function. We use a batch size of 1000, The hidden layer width is the same as the input size. $m_l = 6, m_s = 2$. Learning rate $\eta = 10^{-2}$, a scaling factor is $6 \times 10^{-1}$. We trained 18,000 steps.

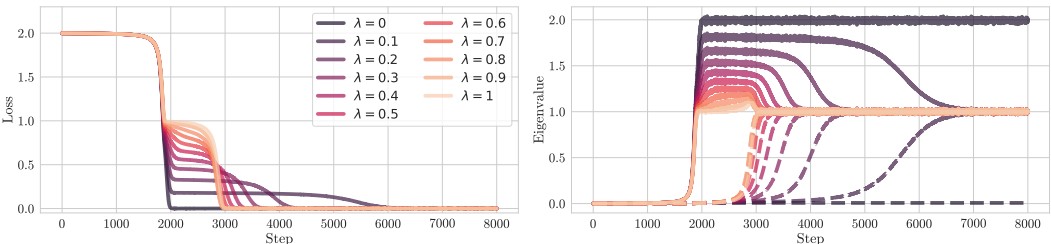

Figure 23: Training loss and eigenvalues of an MLP network with Leaky ReLu as activation on amplitude bias. (Left) Training loss curve showing the relationship between $\lambda$ and convergence behavior. (Right) Corresponding eigenvalue variations demonstrating the impact of the redundancy reduction term.

**Amplitude bias**   We trained a 4-layer mlp with leaky ReLU as activation function. We use a batch size of 1000, The hidden layer width is 8. $c_{ha} = 0.5, c_{la} = 1, f_{ha} = 2\pi/24, f_{la} = 16\pi/24, m = 96$. Learning rate $\eta = 10^{-3}$, a scaling factor is $8 \times 10^{-1}$. We trained 8,000 steps.

## C  GENERALIZATION TO OTHER SSL ALGORITHMS

### C.1  SIMCLR

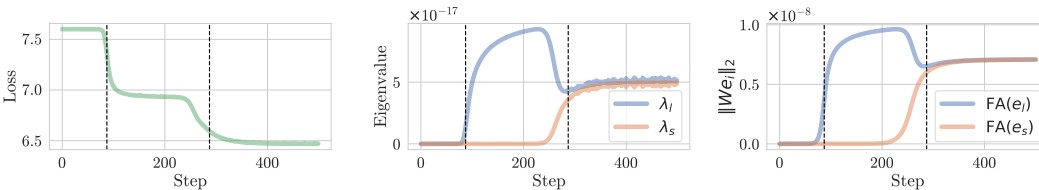

Figure 24: Trained linear layer with batch size as 1000. $m_l = 6, m_s = 2$. Learning rate $\eta = 1 \times 10^{-18}$, a scaling factor is $1 \times 10^{-16}$. We trained 500 steps.

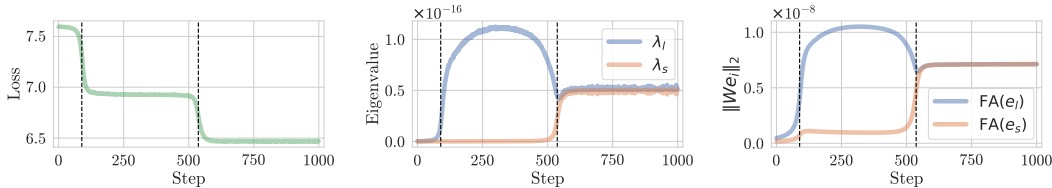

Figure 25: We trained a DLN with 4 layers. The hidden layer width is the same as the input size. The batch size is 1000, $m_l = 6, m_s = 2$. Learning rate $\eta = 5 \times 10^{-3}$, a scaling factor is $6 \times 10^{-1}$. We trained 5,000 steps.

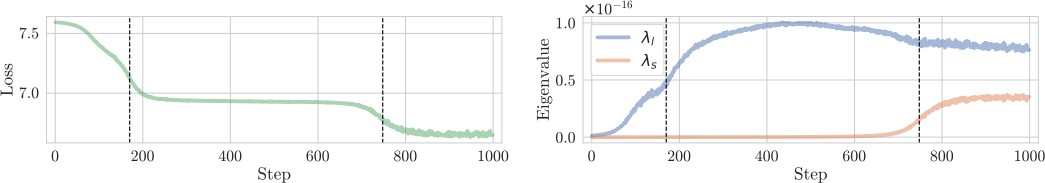

Figure 26: Trained MLP with 4 layer, with LeakyReLU activation. The hidden layer width is the same as the input size. The batch size is 1000, $m_l = 6, m_s = 2$. Learning rate $\eta = 2 \times 10^{-6}$, a scaling factor is $4 \times 10^{-8}$. We trained 5,000 steps.

## C.2 VICREG

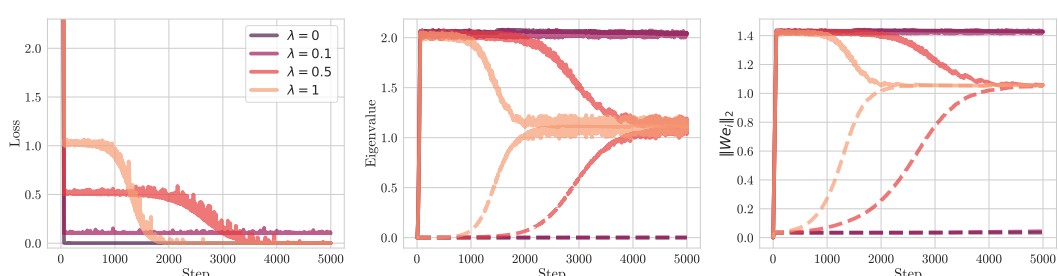

Figure 27: Trained linear layer with batch size as 1000. $m_l = 6, m_s = 2$. Learning rate $\eta = 4 \times 10^{-4}$, a scaling factor is $1 \times 10^{-10}$. We trained 5,000 steps.

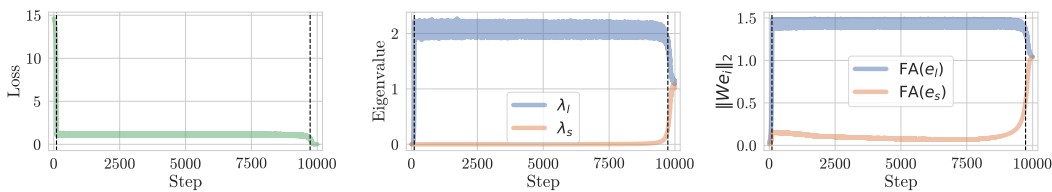

Figure 28: We trained a DLN with 4 layers. The hidden layer width is the same as the input size. The batch size is 1000, $m_l = 6, m_s = 2$. Learning rate $\eta = 9 \times 10^{-4}$, a scaling factor is $3 \times 10^{-1}$. We trained 10,000 steps.

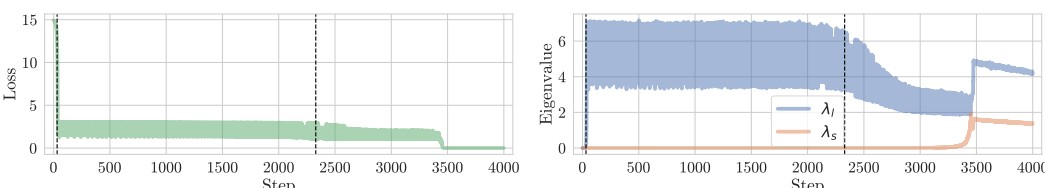

Figure 29: Trained MLP with 4 layer, with LeakyReLU activation. The hidden layer width is the same as the input size. The batch size is 1000, $m_l = 6, m_s = 2$. Learning rate $\eta = 8 \times 10^{-3}$, a scaling factor is $7 \times 10^{-1}$. We trained 5,000 steps.

# D EXPERIMENTAL DETAILS

## D.1 EXTENT BIAS EXPERIMENT

For the extent bias experiment shown in Section 4.1, we train the model using 400 steps. The augmentation noise parameter $a$ was set to 0.01.We use a dataset size of $n = 1000$ samples with feature dimension $m = 10$. We also use a learning rate $\eta = 6 \cdot 10^{-4}$ and a scaling factor $5 \cdot 10^{-1}$.

## D.2 COLORED MNIST EXPERIMENT

For the Colored MNIST shown in Section 7.1, we train the model using default augmentation (RandomResizedCrop[0.7, 1.0], GaussianNoise(sigma=0.1) with randomapply(p=0.5), Normalize) with augmented image size $48 \times 48$. We use background colors as [255, 0, 0], [0, 255,0], [0, 0, 255],[180.31, 180.31, 0], [180.31, 0, 180.31], [0, 180.31, 180.31], [228.07, 114.03, 0], [228.07, 0, 114.03], [114.03, 228.07, 0], [0, 114.03, 228.07]. We trained ResNet-18 with 70 epochs, SGD with learning rate $\eta = 7 \times 10^{-5}$, momentum 0.9, weight decay $10^{-6}$. Batch size is 256, a scaling factor is $10^{-1}$.

## D.3 WATERBIRDS EXPERIMENTS

### D.3.1 MODIFIED WATERBIRDS-C

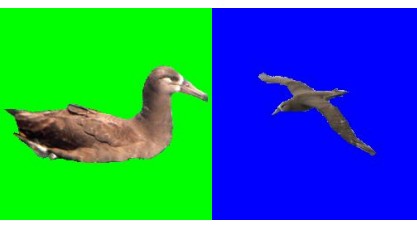

Figure 30: Example of Modified Waterbirds-C dataset.

We train the model using default augmentation (RandomResizedCrop[0.7, 1.0], RandomHorizontalFlip, ColorJitter, Normalize) with augmented image size $96 \times 96$. We trained ResNet-34 with 100 epochs, SGD with learning rate $\eta = 5 \times 10^{-5}$, momentum 0.9, weight decay $10^{-6}$. Batch size is 32, the scaling factor is $2.7 \times 10^{-1}$.

### D.3.2 MODIFIED WATERBIRDS-B

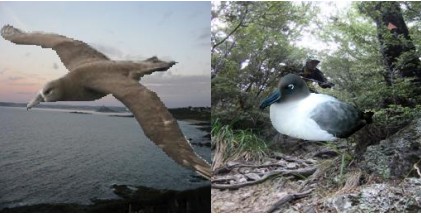

Figure 31: Example of Modified Waterbirds-B dataset.

We train the model using default augmentation (RandomResizedCrop[0.7, 1.0], RandomHorizontalFlip, ColorJitter, Normalize) with augmented image size $96 \times 96$. We trained ResNet-34 with 100 epochs, SGD with learning rate $\eta = 5 \times 10^{-6}$, momentum 0.9, weight decay $10^{-6}$. Batch size is 128, the scaling factor is $7 \times 10^{-2}$.

# E    ADDITIONAL ANALYSES

## E.1    DIFFERENCE IN CRITICAL POINTS BETWEEN LOSS AND EIGENVALUE

The eigenvalue $\lambda_j(\tau) = 1/3$ at $\tau$ (where $\frac{d^2\mathcal{L}}{dt^2}(\tau) = 0$) and $\lambda_j(\tau_j) = 1/2$ (where $\frac{d^2\lambda_j}{dt^2}(\tau_j) = 0$). - For the single-mode loss $\mathcal{L}(t) = (1 - \gamma_j s_j^2(t))^2$, setting the second derivative of the loss to zero at the critical point $\tau$:

$$\frac{d^2\mathcal{L}}{dt^2}\Big|_{t=\tau} = -16\gamma_j \frac{d((1-\lambda_j(t))^2\lambda_j(t))}{dt}\Big|_{t=\tau} = 0$$

which yields $2\lambda_j(\tau) - (1 - \lambda_j(\tau)) = 0$ and $\lambda_j(\tau) = \frac{1}{3}$. According to Theorem 4.2, we have $\lambda_j(\tau_j) = \frac{1}{2}$

## E.2    EIGENVALUES ON SHIFT AUGMENTATION

$$x_{base} = c_a \sin(f_a t + \epsilon_a) + c_b \sin(f_b t + \epsilon_b)$$

$$\epsilon_a, \epsilon_b \overset{\text{i.i.d.}}{\sim} U(-\pi, \pi)$$

$$\Gamma = \mathbb{E}[x_{base} x_{base}^\top]$$

$$\Gamma_{ij} = \mathbb{E}[c_a^2 \sin(f_a i + \epsilon_a)\sin(f_a j + \epsilon_a) + c_a c_b \sin(f_a i + \epsilon_a)\sin(f_b j + \epsilon_b)$$

$$+ c_a c_b \sin(f_b i + \epsilon_b)\sin(f_a j + \epsilon_a) + c_b^2 \sin(f_b i + \epsilon_b)\sin(f_b j + \epsilon_b)]$$

$$\begin{aligned}
\mathbb{E}_{\epsilon_a,\epsilon_b}[\sin(\theta_a + \epsilon_a)\sin(\theta_b + \epsilon_b)] &= \mathbb{E}_{\epsilon_a,\epsilon_b}[\text{Im}(\exp(i(\theta_a + \epsilon_a)))\,\text{Im}(\exp(i(\theta_b + \epsilon_b)))] \\
&= \mathbb{E}_{\epsilon_a}[\text{Im}(\exp(i(\theta_a + \epsilon_a)))]\mathbb{E}_{\epsilon_b}[\text{Im}(\exp(i(\theta_b + \epsilon_b)))] \\
&= \text{Im}(\mathbb{E}_{\epsilon_a}[\exp(i(\theta_a + \epsilon_a))])\,\text{Im}(\mathbb{E}_{\epsilon_b}[\exp(i(\theta_b + \epsilon_b))]) \\
&= \text{Im}(\mathbb{E}_{\epsilon_a}[\exp(i\epsilon_a)\exp(i\theta_a)])\,\text{Im}(\mathbb{E}_{\epsilon_b}[\exp(i\epsilon_b)\exp(i\theta_b)]) \\
&= \text{Im}(\varphi(1)\exp(i\theta_a))\,\text{Im}(\varphi(1)\exp(i\theta_b))
\end{aligned}$$

We can define $u, d$ as $u = \mu + \alpha, d = \mu - \alpha, \alpha = 2\pi$.

$$\varphi(1) = \frac{\exp(iu) - \exp(id)}{i(u - d)} = \frac{\exp(i\mu)}{\alpha i}\frac{\exp(i\alpha) - \exp(-i\alpha)}{2i} = \frac{\exp(i\mu)}{\alpha i}\sin(\alpha) = 0$$

So,

$$\mathbb{E}_{\epsilon_a,\epsilon_b}[\sin(\theta_a + \epsilon_a)\sin(\theta_b + \epsilon_b)] = 0$$

Similar,

$$\begin{aligned}
\mathbb{E}[\sin(\theta_a + \epsilon_a)\sin(\theta_b + \epsilon_a)] &= -\frac{1}{2}\mathbb{E}[\cos(\theta_a + \theta_b + 2\epsilon_a) - \cos(\theta_a - \theta_b)] \\
&= -\frac{1}{2}\mathbb{E}[\cos(\theta_a + \theta_b + 2\epsilon_a)] + \frac{1}{2}\cos(\theta_a - \theta_b) \\
&= -\frac{1}{2}\int_a^b [\frac{1}{b-a}\cos(\theta_a + \theta_b + 2x)dx] + \frac{1}{2}\cos(\theta_a - \theta_b) \\
&= -\frac{1}{4}\frac{1}{b-a}[\sin(\theta_a + \theta_b + 2b) - \sin(\theta_a + \theta_b + 2a)] + \frac{1}{2}\cos(\theta_a - \theta_b) \\
&= -\frac{1}{4}\frac{1}{b-a}[2\cos(\theta_a + \theta_b + a + b)\sin(b - a)] + \frac{1}{2}\cos(\theta_a - \theta_b)
\end{aligned}$$

we assumed $b - a = 2\pi$,

$$\mathbb{E}[\sin(\theta_a + \epsilon_a)\sin(\theta_b + \epsilon_a)] = \frac{1}{2}\cos(\theta_a - \theta_b)$$

finally, we get

$$\Gamma_{ij} = \frac{c_a^2}{2}\cos(f_a(i-j)) + \frac{c_b^2}{2}\cos(f_b(i-j))$$

is a symmetric circulant matrix when $f_a = a\frac{2\pi}{N}, f_b = b\frac{2\pi}{N}$,

$$c_j = \frac{c_a^2}{2}\cos(f_a j) + \frac{c_b^2}{2}\cos(f_b j)$$

$$\Lambda_{\Gamma,k} = \sum_{j=0}^{N-1} c_j \omega^{-kj}$$

$$V_{\Gamma,k} = \frac{1}{\sqrt{N}}\left[1, \omega^k, \omega^{2k}, \ldots, \omega^{(N-1)k}\right]^\top$$

$$\omega = \exp(\frac{2\pi i}{n}) = \cos(\frac{2\pi}{n}) + i\sin(\frac{2\pi}{n})$$

This is symmetric, so eigenvalues are real. The eigenvectors can be expressed either in complex form or as pairs of real vectors. Using properties of Discrete Fourier Transform (DFT) matrix on $\Lambda_{\Gamma,k}$,

$$\Lambda_{\Gamma,k} = \begin{cases} 0 \ (k \neq l_a, N - l_a, l_b, N - l_b) \\ \frac{c_a^2}{2} \ (k = l_a \text{ or } k = N - l_a) \\ \frac{c_b^2}{2} \ (k = l_b \text{ or } k = N - l_b) \end{cases}$$

Finally, we can derive as:

$$\Lambda_\Gamma = \text{diag}\left(\left[\frac{c_a^2}{2}, \frac{c_a^2}{2}, \frac{c_b^2}{2}, \frac{c_b^2}{2}, \mathbf{0}_{m-2}\right]\right),$$

$$V_\Gamma^{(\leq 4)} = \left[\frac{1}{\sqrt{N}}e_{h,\cos} \ \frac{1}{\sqrt{N}}e_{h,\sin} \ \frac{1}{\sqrt{N}}e_{l,\cos} \ \frac{1}{\sqrt{N}}e_{l,\sin}\right].$$

where

$$e_{h,\cos} = c_a \cos(f_a t),$$
$$e_{h,\sin} = c_a \sin(f_a t),$$
$$e_{l,\cos} = c_b \cos(f_b t),$$
$$e_{l,\sin} = c_b \sin(f_b t).$$

### E.3 ABLATION STUDIES ON LEAKY RELU

We confirmed that nonlinearity has the effect of reducing the initial eigenvalue $\lambda$ by varying the slope of the leaky ReLU activation function. Figure 32 shows as the slope approaches zero, the function converges to ReLU, and it can be observed that $\tau$ exhibits slower dynamics.

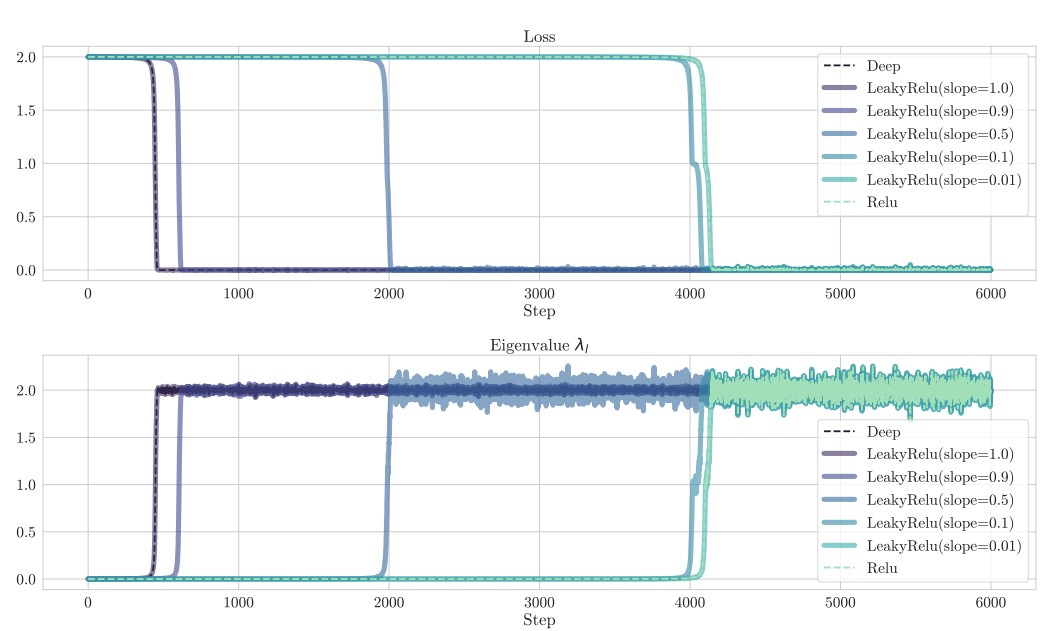

Figure 32: Variation of loss and eigenvalue $\lambda_l$ as a function of Leaky ReLU slope parameter on $\lambda = 0$ Barlow Twins Loss.

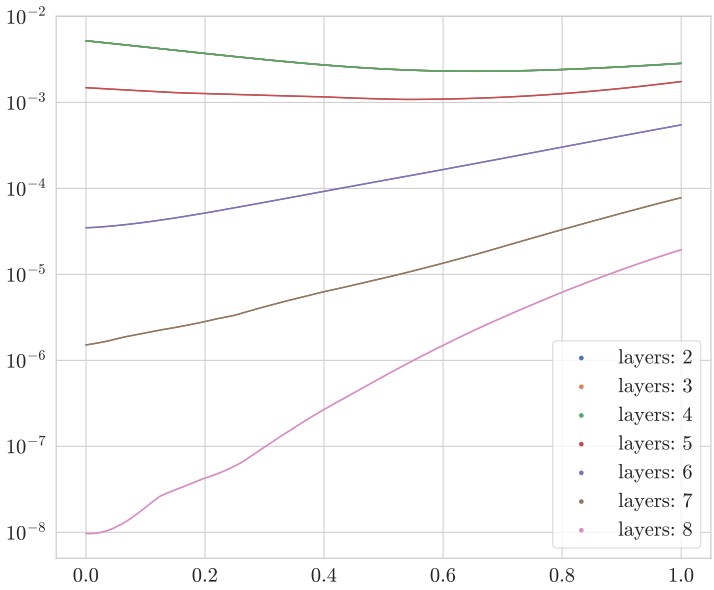

Figure 33: Initial top eigenvalue of correlation matrix $C$ as a function of Leaky ReLU slope parameter. A slope of 0 corresponds to ReLU activation, while a slope of 1 represents the Deep Linear Network case. As the number of layers increases, the reduction in initial eigenvalue $\lambda$ due to decreasing slope leads to slower learning dynamics.

## E.4 BATCHNORM EFFECTS

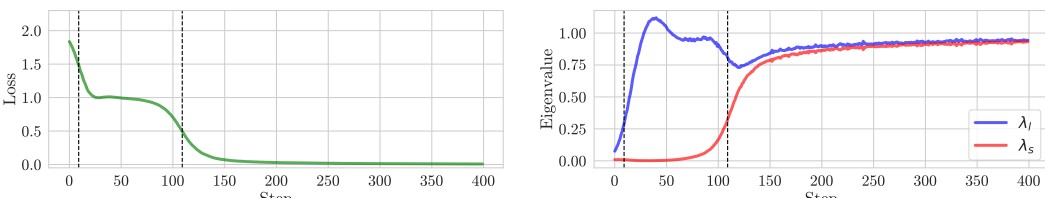

Figure 34: **Effects of batch normalization on learning dynamics in** $\lambda = 1$ **non-linear network.** (Left) Stepwise learning curves of Barlow Twins showing two distinct learning phases with vertical dashed lines marking critical transition points during training. The green line shows empirical loss decreasing in two clear stages. (Right) Evolution of eigenvalues $\lambda_j$ of correlation matrix C during training. The eigenvalue $\lambda_l$ (blue) increases first, followed by the eigenvalue $\lambda_s$ (red). We use same inputs in Figure 1.

## E.5 RANDOM RESIZED CROP EFFECTS

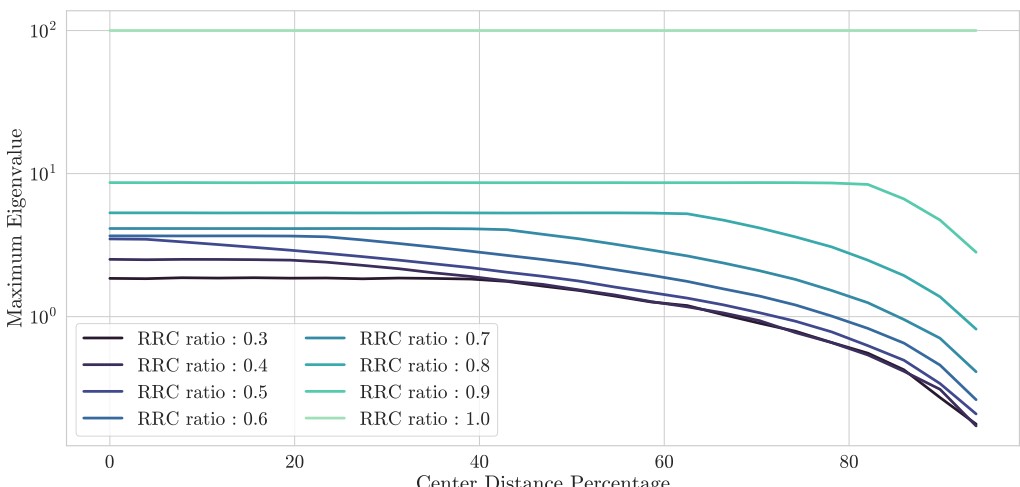

Figure 35: **Effect of random resized crop (RRC) on the eigenvalues of the feature cross-correlation matrix in a toy extent setting.** We consider $x_{\text{base}} = [0, \cdots, 0, b\mathbf{1}_{m_b}, 0, \cdots, 0] \in \mathbb{R}^m$, where $m_b = 20, m = 400$, containing a single non-zero feature block (all other pixels are zero), and compute the cross-correlation matrix $\Gamma$ of two RRC-augmented views. The curves show the largest eigenvalue $\gamma_{\max}(\Gamma)$ (log scale) as a function of the feature distance from the center (%), for different minimum crop scales $r \in \{0.3, \ldots, 1.0\}$ (lighter colors indicate larger $r$). When cropping is not applied ($r = 1.0$), $\gamma_{\max}$ is constant across positions, while under RRC it monotonically decreases as the feature moves towards the border, with stronger decay for more aggressive crops (smaller $r$). This illustrates that RRC induces a *center bias* by assigning larger eigenvalues to central features and suppressing peripheral ones.

We investigate the impact of Random Resized Crop (RRC) on feature learning, specifically focusing on the spatial bias it induces. First, we analyze the effect of RRC from a theoretical perspective. As shown in Figure 35, the largest eigenvalue of the feature cross-correlation matrix monotonically decreases as the feature location moves away from the center. This decay is more pronounced under aggressive cropping strategies (smaller minimum scale $r$), indicating that RRC inherently suppresses peripheral features while enhancing central ones. To empirically validate this spatial bias, we designed a synthetic dataset consisting of two independent components, as illustrated in Figure 36: a center object (circle) and an outer object (square). This setup allows us to decouple the spatial location from semantic content.

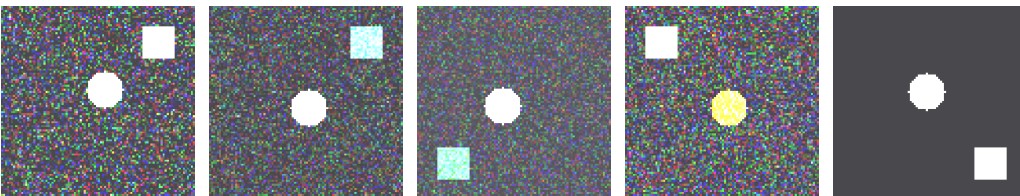

Figure 36: **Overview of the RRC effect Dataset.** Each image ($96 \times 96$) consists of two independent components: a center object (circle) and an outer object (square). The center object serves as the primary classification target, **represented as a circle with an area approximately equivalent to the $16 \times 16$ square**, sampled from 10 distinct color classes. The outer object is a $16 \times 16$ square placed in one of the four corners, with a color independently drawn from a separate set of 10 classes.

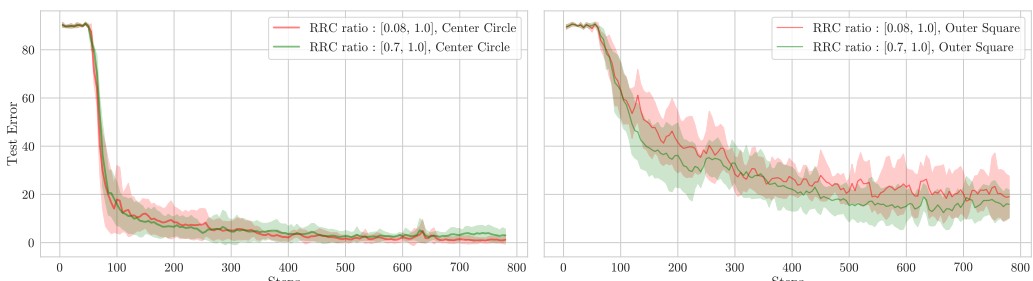

Figure 37: **Impact of Random Resized Crop (RRC) scale on the learnability of center and outer objects.** The left and right panels show the **color classification** test error for the center circle and the outer square, respectively (same y-axis scale). **All models were trained using the Barlow Twins objective with a batch size of 32.** We observe three key trends: (1) The center object consistently achieves lower test error compared to the outer object due to the spatial nature of RRC. (2) For the center object, aggressive cropping (ratio $[0.08, 1.0]$) induces a stronger center bias, leading to marginally improved accuracy. (3) Conversely, for the outer object, a milder cropping strategy (ratio $[0.7, 1.0]$) mitigates this center bias, resulting in significantly better performance compared to the aggressive setting.

The experimental results in Figure 37 align with our theoretical analysis. We observe distinct learning dynamics depending on the object's location and the crop scale. The model consistently achieves lower test errors on the center object compared to the outer object. Notably, aggressive cropping (ratio $[0.08, 1.0]$) exacerbates the center bias, improving performance on the center object but degrading it on the outer object. Conversely, a milder cropping strategy (ratio $[0.7, 1.0]$) mitigates this bias, allowing the model to learn peripheral features more effectively.

## E.6 AUGMENTED BACKGROUND MODIFIED WATERBIRDS-B

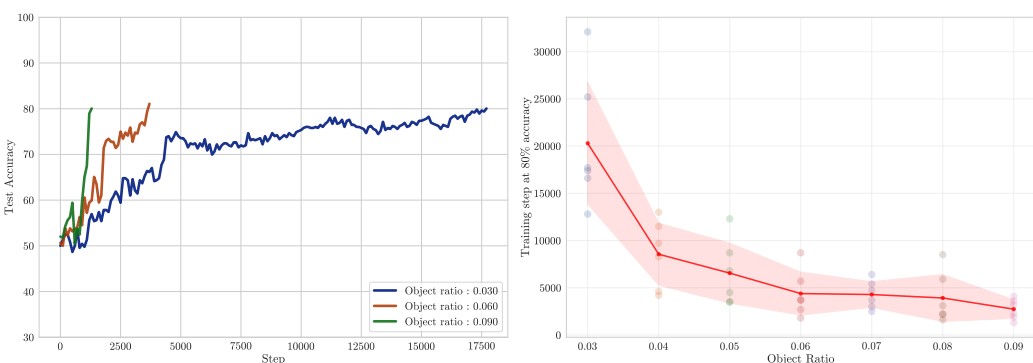

Figure 38: **Effect of object extent on learning speed without spurious correlations (Modified Waterbirds-B).** We use a variant of MODIFIED WATERBIRDS-B where the natural backgrounds are independent of the labels (no spurious correlation): background images are resized from 256 to 192 pixels and then randomly resized-cropped to $144 \times 144$. For each object ratio (fraction of image area occupied by the bird), we train Barlow Twins on this dataset and record the training step at which $80\%$ training accuracy is first reached. Colored dots show individual random seeds, the red curve shows the mean over seeds, and the red shaded region indicates one standard deviation. As the object ratio increases, the number of steps needed to reach $80\%$ accuracy consistently decreases, indicating that larger object extent accelerates learning even when backgrounds do not carry spurious signal.

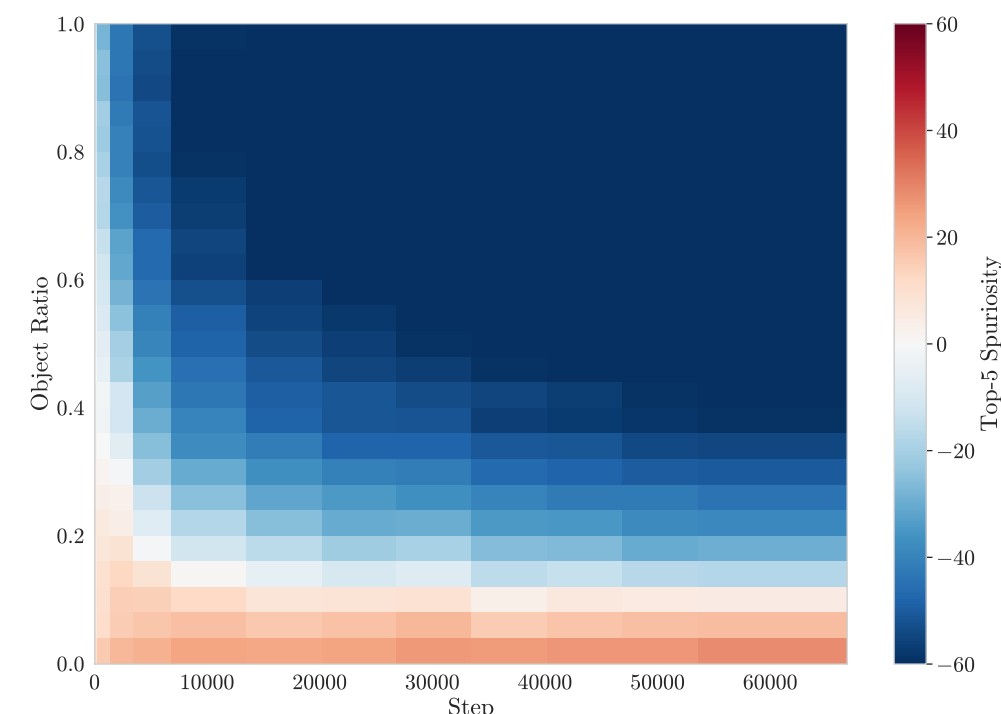

Figure 39: **Evolution of spurious correlations during training on ImageNet validation set.**
We visualize how spuriosity changes across training steps and object ratios. We define spuriosity
as Spuriosity $=$ Acc$_{\text{only bg}}$ $-$ Acc$_{\text{only object}}$, where Acc$_{\text{only bg}}$ is the top-5 accuracy when only the
background is visible, and Acc$_{\text{only object}}$ is the top-5 accuracy when only the object is visible. The
object ratio represents the proportion of the image occupied by the bounding box. Positive spuriosity
(red) indicates the model relies more on background features, while negative spuriosity (blue) indicates
reliance on object features. Each row represents a bin of samples with similar object ratios.

### E.7 Object Ratio with Spurious Feature Learning on ImageNet

To understand how models learn spurious correlations during training, we train a ResNet-50 on
ImageNet and evaluate it using linear probing at various checkpoints. We use linear probing to freeze
the pretrained feature extractor and train only a linear classifier on top, assessing what features
the backbone network has learned without fine-tuning. The heatmap in Figure 39 reveals a clear
relationship between object ratio and spurious feature learning during training on the ImageNet
validation set. Samples with smaller objects (lower object ratios, bottom rows) exhibit positive
spuriosity, indicating that the model learns to rely heavily on background features for classification.
This background dependence increases as training progresses, as evidenced by the deepening red color
in the lower region. In contrast, samples with larger objects (higher object ratios, top rows) maintain
negative spuriosity throughout training, demonstrating that the model correctly learns object-centric
features when objects occupy a substantial portion of the image. The boundary between positive and
negative spuriosity (white region around object ratio $\approx 0.2$) remains relatively stable across training
steps, suggesting that object ratio acts as a consistent factor in determining whether a model learns
spurious background or causal object.

## F  Limitations

Our study has several limitations due to its simplified assumptions. While our theoretical analysis
provides valuable insights into the relationship between extent bias and shortcut learning, several
limitations should be acknowledged:

- Feature Independence: Our assumption of independent features may not reflect complex interdependencies in practical scenarios.
- Augmentation Limitations: Our basic augmentation approach may not fully represent the sophisticated strategies used in modern SSL methods.

Future work could address these limitations by extending the theoretical framework incorporating feature interactions and analyzing the impact of more complex augmentation strategies.

## G   LLM USAGE

LLMs were used solely for language editing (grammar, spelling, and style improvements) and code generating, but not for generating research ideas, experimental design, or scientific content. All LLM-generated content was verified and validated by the authors.

