# OpenReview forum: "Stepwise Feature Learning in Self-Supervised Learning"
_ICLR.cc/2026/Conference — Submitted to ICLR 2026_

### Official Review · Reviewer_Q3Bm · 2025-10-30

**Soundness:** 2
**Presentation:** 2
**Contribution:** 1
**Rating:** 2
**Confidence:** 3

**Summary:**

The paper analyzes stepwise feature learning dynamics in SSL through the lens of feature cross-correlation eigenvalues. It introduces the notions of extent bias and amplitude bias. It provides theoretical derivations on toy examples, and experiments on synthetic and semi-synthetic data.

**Strengths:**

- The paper provides a theoretical exposition of stepwise feature learning in SSL, connecting it to short cut learning.

- In addition to the theoretical analysis, the paper presents synthetic experiments in more general settings as well as evaluations on semi-synthetic image datasets.

**Weaknesses:**

- The definition of extent bias is conceptually vague. The theoretical example in line 160 does not clearly distinguish “extent” from “amplitude”. The example effectively has just two 1-dimensional features (along different directions), but different magnitudes (one of size $\sqrt{m_l}$ and the other of size $\sqrt{m_s}$). Hence the analysis effectively reduces to showing that features with larger magnitudes are learned earlier.

- Following the previous point, both examples essentially show that features with larger magnitudes are learned earlier which is rather obvious, while offering little additional insight.

- Overall, I don’t see what new insights or significant practical implications the paper offers. From the theoretical side, the claimed novelty over Simon et al. (2023) is limited. The paper largely shows the same stepwise dynamics for SSL but just on two specific toy examples, without introducing fundamentally new mechanisms or insights. Additionally, theoretical analysis on how features with larger magnitudes and higher dimensions can suppress the learning of other features in SSL already appears in earlier works such as [1]. On the experimental side, the results only confirm the stepwise phenomenon. It’s unclear what specific new guidance for practice can be drawn from this study. Overall, the work seems to lack theoretical depth, novelty, and practical significance.

[1] Xue, Yihao, et al. "Which features are learnt by contrastive learning? on the role of simplicity bias in class collapse and feature suppression." International Conference on Machine Learning. PMLR, 2023.

**Questions:**

Please see the questions raised in the Weaknesses section.

---

> ### Author Response · Authors · 2025-11-21
>
> We sincerely thank the reviewer for the detailed feedback. We address each point below.
>
> ---
>
> > Overall, I don't see what new insights or significant practical implications the paper offers. From the theoretical side, the claimed novelty over Simon et al. (2023) is limited. The paper largely shows the same stepwise dynamics for SSL but just on two specific toy examples, without introducing fundamentally new mechanisms or insights.
>
> We focus on **(1) which features are learned first, (2) how realistic loss dynamics behave, and (3) how these dynamics connect to shortcut learning**, rather than merely confirming that stepwise behavior occurs.
>
> While Simon et al. (2023) established the fundamental *existence* of stepwise learning, our work identifies the *determinants* of this order and links them to practical shortcut learning phenomena. The explicit differences are summarized below:
>
> | Aspect | Simon et al. (2023) | **Ours** |
> | :--- | :--- | :--- |
> | **1-1. Core Question** | **Why** does SSL exhibit stepwise learning? (Focus on existence) | **Which** features are learned first? (Focus on determinants: Extent & Amplitude) |
> | **1-2. Feature Semantics** | Eigenvectors remain mathematical "directions" **without a semantic** interpretation. | Eigenvectors are **linked to physical structures** (Background vs. Object, Low vs. High Freq). |
> | **2-1. Loss Dynamics** | Analyzed **only linear Barlow Twins with $\lambda=1$** (independent learning). | Analyzed **general redundancy reduction term ($0 \le \lambda \le 1$)** (Section 6.1, A.5). |
> | **2-2. Architecture Scope** | Primarily **Linear** models. | Verified across **Linear** (Figs. 6, 7), **Deep Linear** (Figs. 20, 21), and **MLPs** (Figs. 22, 23). |
> | **3-1. Implication** | Theoretical observation of learning phases. | **Causal link to Shortcut Learning.** Identifies a mechanism that naturally suggests potential mitigation strategies ( $\lambda$-tuning, data preprocessing, additional analysis on augmentation), which we plan to articulate more explicitly and evaluate empirically in a revised version of the paper. |
>
>
>
> **Detailed Breakdown of Contributions:**
>
> 1-1. **Core Question**: While Simon et al. focused on the fundamental question of **why** SSL exhibits stepwise learning, our work specifically identifies **which types of features** are learned at each stage of this structure and establishes a direct connection to the phenomenon of shortcut learning.
>
> 1-2. **Connecting Eigenvectors to Interpretable Structures**: Unlike previous work where eigenvectors remained abstract, we explicitly map them to **Extent Bias** (background vs. object) and **Amplitude Bias** (low vs. high frequency), providing a concrete mechanism for *why* shortcuts are learned first.
>
> 2-1. **Generalization to redundancy reduction term $\lambda \neq 1$**: We extend the analysis beyond the independent learning case ($\lambda=1$). We show that in the general case ($0 < \lambda < 1$), features interact through the redundancy reduction term $\lambda$, meaning new modes can be suppressed or can adjust earlier ones.
>
> 2-2. **Robustness across Architectures**: We rigorously verify that these stepwise dynamics are not artifacts of shallow networks but persist in Deep Linear Networks and MLPs, confirming their relevance to realistic deep learning scenarios.
>
> 3-1. **Practical Implications for Shortcut Learning**: We provide detailed analyses in the discussion below. While Simon et al. demonstrated that stepwise patterns appear in methods like Barlow Twins, SimCLR, and VICReg, we interpret these shortcuts through the mechanism of plateau reduction in controlled datasets such as Colored MNIST and Modified Waterbirds. Furthermore, we derive actionable guidance for data augmentation, demonstrating that increasing the object extent relative to the background significantly shortens the phase dominated by shortcut learning.

---

> ### Author Response · Authors · 2025-11-21
>
> > Theoretical analysis on how features with larger magnitudes and higher dimensions can suppress the learning of other features in SSL already appears in earlier works such as [1]. On the experimental side, the results only confirm the stepwise phenomenon. It's unclear what specific new guidance for practice can be drawn from this study. Overall, the work seems to lack theoretical depth, novelty, and practical significance.
>
> The fundamental differences are summarized below:
>
> | Aspect | Xue et al. [1] (ICML 2023) | **Ours** |
> | :--- | :--- | :--- |
> | **Target Framework** | **Supervised / Unsupervised Contrastive Learning** Focuses on contrastive objectives with positive/negative pairs, primarily in settings where labels or subclass structure guide which similarities are encouraged. | **Redundancy Reduction SSL** Studies Barlow Twins–style objectives that match cross-correlations between two augmented views, without explicit negatives or label supervision, emphasizing decorrelation and variance preservation across feature dimensions. |
> | **Theoretical Basis** | **Implicit Bias of SGD** towards Min-Norm Solution. Analyzes how gradient descent in (linearized) contrastive models converges to the minimum-ℓ₂-norm interpolating solution, and how this implicit bias determines which features are retained or discarded. | **Eigenvalue Decomposition** of Feature Cross-Correlation Matrix. Derives closed-form dynamics by diagonalizing the cross-correlation matrix into eigenmodes with eigenvalues $\gamma_j$. Each eigenmode evolves independently, and its eigenvalue directly governs both the learning rate and steady-state magnitude of the corresponding feature. |
> | **Feature Interaction** | **Competition & Suppression:** "Easy" features dominate, causing "hard" features to be **unlearned** or ignored to minimize the norm. | **Sequential Prioritization:** Features **co-exist** but are learned at different times. Large features are learned *first*, followed by smaller ones. |
> | **Capacity Constraint** | Often assumes **Dimensional Bottleneck** (limited embedding size forces selection). When representation dimension is small or augmentations frequently erase class-relevant information, the model is forced to choose a subset of features that minimize the norm, leading to feature competition. | Occurs **even with sufficient embedding dimension**, driven purely by the spectrum of $\gamma_j$. Our analysis shows that the bias toward large-extent/amplitude features emerges **without** architectural or dimensional bottlenecks. |
> | **Key Phenomenon** | **Class Collapse / Feature Collapse:** Explains how models may converge to embeddings that over-emphasize a few “easy” or spurious directions, effectively losing diversity in the learned features (permanent or long-lasting suppression of certain subclass-relevant features). | **Stepwise Dynamics of Feature Learning:** Explains *when* different features emerge during training. The apparent “shortcut” behavior arises because large features are learned **first**, but smaller features can still be acquired later. Thus, the issue is one of **learning priority**, not inevitable or permanent feature suppression. |
>
> While both papers identify "large features" as the cause, **the mechanisms and mathematical structures differ fundamentally**.
> In other words, [1] explains which features remain in the min-norm solution of contrastive objectives under bottlenecks, while we characterize the temporal order of feature learning in redundancy-reduction SSL without such bottlenecks.
>
> ---
>
> > The definition of extent bias is conceptually vague. The theoretical example in line 160 does not clearly distinguish "extent" from "amplitude". The example effectively has just two 1-dimensional features (along different directions), but different magnitudes (one of size  and the other of size ). Hence the analysis effectively reduces to showing that features with larger magnitudes are learned earlier.
>
> Our analysis does **not simply** reduce to showing that "features with larger magnitudes are learned earlier". Extent and amplitude represent two distinct and naturally separable axes in image features: (1) spatial coverage (how many pixels a feature spans) and (2) per-pixel intensity. We explicitly distinguish these concepts:
>
> - Extent bias: Keeping per-coordinate amplitude fixed while varying spatial coverage
> - Amplitude bias: Keeping spatial coverage fixed while varying per-coordinate amplitude
>
> We will clarify this distinction more explicitly in the revised manuscript with enhanced mathematical definitions.

---

> ### Author Response · Authors · 2025-11-21
>
> > Both examples essentially show that features with larger magnitudes are learned earlier which is rather obvious, while offering little additional insight.
>
> While it might seem intuitive that "larger magnitudes are learned earlier," this is not a general law of neural network learning dynamics; it depends critically on the objective, architecture, and data structure.
>
> In supervised learning, particularly in regression, magnitude does not determine learning order:
>
> - Regression: Spectral bias research [2]: In the function $\sum_i A_i \sin(2\pi k_i z + \phi_i)$, low-frequency features are learned first regardless of whether $A_i = 1$ (uniform amplitude) or $A_i$ is monotonically increasing [2][3].
> - Classification [4]: Networks learn "easily distinguishable" frequencies first, not necessarily high-amplitude ones.
> - CNN texture bias [5]: CNNs rely on local texture/patches rather than global shape, independent of feature magnitude.
>
> These results demonstrate that learning order is not simply determined by norm or extent, but varies by objective, architecture, and data structure. In this context, **our contribution is showing that in redundancy-reduction-based "SSL", extent and amplitude critically determine the critical time** with closed-form characterizations, and connecting this to shortcut learning. Moreover, we verify that these extent/amplitude biases persist in deep linear networks as shown in Fig. 20 and Fig. 21 (not just single-layer toys as shown in Fig. 1 and Fig. 9), confirming these phenomena are stable in deeper architectures.
>
> ---
>
> > It's unclear what specific new guidance for practice can be drawn from this study.
>
> Concretely, our analysis suggests three practical strategies to **mitigate shortcut learning and encourage generalizable features** for SSL practitioners:
>
> 1. **$\lambda$-tuning (decorrelation strength)**
> Our general Barlow Twins analysis for $0 \le \lambda \le 1$ shows that appropriately increasing the redundancy reduction term reduces the dominance of (potentially spurious) large-extent/amplitude features and facilitates learning of other more predictive features. This provides a principled guideline for setting $\lambda$ beyond heuristic tuning aimed purely at decorrelation.
>
> 2. **Data preprocessing design (object–background extent)**
>    Our Colored MNIST and Modified Waterbirds experiments demonstrate that enlarging object regions or using crops that reduce background extent shortens the shortcut-dominated plateau. This suggests a concrete data/preprocessing strategy: when possible, increase the object-to-background ratio to mitigate shortcut learning driven by background features.
>
> 3. **Data augmentation design (extent bias via RRC)**
> Our analysis yields a simple design guideline: **by adjusting the aggressiveness of Random Resized Crop (RRC), one can regulate the trade-off between rapidly learning central targets and preserving peripheral features**. Using aggressive RRC can induce a "center bias"—a tendency where the model relies primarily on the central region of an image. This phenomenon stems from the theoretical definition of $\Gamma$ as an expectation over augmented pairs $(x, x')$; modifying the cropping pipeline directly reshapes the spectral properties of $\Gamma$. We focused on RRC, as recent work [6] on Joint Embedding Architectures (JEA) suggests that strong performance can be achieved using essentially only random cropping. As detailed in our controlled experiments (Appendix E.5), RRC acts as a spatial filter. Fig. 35 empirically demonstrates this mechanism: the maximum eigenvalue $\gamma_{max}$ monotonically decreases as a feature moves away from the image center, and this decay becomes significantly steeper under aggressive cropping (lower minimum scale $r$, e.g., 0.3), thereby suppressing the eigenvalues of peripheral features. To validate this effect on learning dynamics, we utilized a synthetic dataset containing semantically independent 'Center Circle' and 'Outer Square', allowing us to decouple spatial location from semantic content (Fig. 37). The resulting performance dynamics confirm aggressive cropping (ratio $[0.08, 1.0]$) amplifies the center bias, rapidly minimizing error for the center object while degrading performance on the outer object. Conversely, a milder cropping strategy (ratio $[0.7, 1.0]$) mitigates this suppression, allowing the model to effectively learn the outer object. These findings yield design guidelines that adjusting crop aggressiveness regulates the trade-off between learning central targets and peripheral features.
>
>
>
> We will highlight these practical implications more clearly in the revised manuscript.

---

> ### Author Response · Authors · 2025-11-21
>
> ---
> ### References
> [1] Xue, Yihao, et al. "Which features are learnt by contrastive learning? on the role of simplicity bias in class collapse and feature suppression." International Conference on Machine Learning. PMLR, 2023.
>
> [2] Rahaman, Nasim, et al. "On the spectral bias of neural networks." International conference on machine learning. PMLR, 2019.
>
> [3] Basri, Ronen, et al. "The convergence rate of neural networks for learned functions of different frequencies." Advances in Neural Information Processing Systems, 2019.
>
> [4] Wang, Shunxin, et al. "What do neural networks learn in image classification? A frequency shortcut perspective." Proceedings of the IEEE/CVF International Conference on Computer Vision, 2023.
>
> [5] Geirhos, Robert, et al. "ImageNet-trained CNNs are biased towards texture; increasing shape bias improves accuracy and robustness." International conference on learning representations, 2018.
>
> [6] Moutakanni, Théo, et al. "You don’t need domain-specific data augmentations when scaling self-supervised learning." Neural Information Processing Systems, 2024.

---

### Official Review · Reviewer_cfsZ · 2025-10-31

**Soundness:** 3
**Presentation:** 3
**Contribution:** 2
**Rating:** 6
**Confidence:** 4

**Summary:**

The paper analyzes self-supervised learning (SSL) algorithms through the theoretical framework introduced by Simon et al. (2023). This framework employs simplified toy one-layer linear models to examine the behavior of SSL objectives—particularly the Barlow Twins (BT) loss.

Within this setup, the authors first explore extent bias, aiming to understand how feature learning in SSL is influenced more by dimensional properties than by semantic content. They design a toy dataset in which each input is a concatenation of constant vectors of varying dimensionalities, randomly modulated in amplitude. Using the analytical tools developed by Simon et al. (2023), they study the temporal learning dynamics and show that the dimensional scale plays a key role in shaping learning behavior.

The second part of the study investigates amplitude bias, referring to the tendency of networks to favor low-frequency information during training. In this case, the inputs are modeled as superpositions of randomly weighted high- and low-frequency cosine signals. Applying the same analytical framework, the authors demonstrate that the learning dynamics depend primarily on the amplitude of spectral components rather than their frequency.

In the following sections, the paper examines the effect of the redundancy reduction coefficient of BT loss and elaborates on the extension of  the analysis to nonlinear networks. Finally, numerical experiments on the Colored MNIST and Waterbirds datasets are presented to support the theoretical findings.

**Strengths:**

- The authors present clear problem statements and theorems, supported by sound proofs in the Appendix, for both extent bias and amplitude bias. These analyses, based on linear toy models, are insightful and convincing. The fact that the findings are further validated through experiments on real data enhances their value.

- The article is well-presented and easy, even pleasant, to read.

- Overall, this work represents a valuable extension of prior research. Both extent bias and frequency (amplitude) bias have been recognized in the literature as important issues, and this paper provides additional insights by applying the analytical framework of Simon et al. (2023).

**Weaknesses:**

- The main limitation lies in the simplicity of the toy linear or single-layer model used in the analysis.

- More comprehensive experiments with real-world datasets would further strengthen the empirical validity of the presented analysis results.

**Questions:**

- What is the dependence of the $\Gamma$ matrix on the chosen data augmentations?

- Could you elaborate on the generalizability of the observations to other SSL algorithms? The derivations are based on the Barlow Twins loss, but can the underlying intuitions about common features of algorithms/losses to extend results to other losses or algorithmic designs?

- Regarding the extent vs. semantic impact: would it be possible to conduct controlled experiments with manipulated data (e.g., keeping the objects fixed while modifying the background—by expanding, changing texture, etc.) to test the influence of these properties?

---

> ### Author Response · Authors · 2025-11-21
>
> We thank the reviewer for the thoughtful and positive assessment of our work. We address each point below.
>
> > W1 : The main limitation lies in the simplicity of the toy linear or single-layer model used in the analysis.
>
> Our analysis uses the toy linear or single-layer model as an analytically tractable model, and these predictions are then **validated in deep and non-linear networks** so that our conclusions do not rely solely on the toy setting.
>
> To support the relevance of our findings beyond single-layer linear models, we complement the theory with:
>
> - **Deep linear networks (DLNs).**
>   In Section 6.2 and Appendix B.2, we analyze deep linear networks and show that, although depth induces non-linear learning dynamics, the eigenvalue-driven ordering persists: features associated with larger eigenvalues of $\Gamma$ are still learned earlier.
>
> - **Non-linear networks (MLPs).**
>   In Section 6.3 and Appendix B.3, MLPs with ReLU/Leaky ReLU nonlinearities exhibit stepwise learning dynamics consistent with our theory: high-extent or high-amplitude features are learned earlier. In addition, we include an ablation study in Appendix E.3 demonstrating that the eigenvalues of the cross-correlation matrix $C$ decrease as the degree of nonlinearity increases.
>
> - **Realistic deep CNN architectures.**
>   In Section 7, ResNet-18/34 experiments on hybrid real–synthetic datasets (Colored MNIST and Modified Waterbirds) show extent bias, plateau phenomena, and object/background-extent dependence in non-linear convolutional settings, again matching the predictions from our linear analysis.
>
> We have clarified this “toy model as lens + deep/non-linear validation” narrative in the revised introduction and Section 6, making explicit that the toy analysis provides a foundation whose key qualitative prediction—eigenvalue-based ordering of which features are learned earlier—is observed in more realistic architectures as well.
>
>
> > W2 : More comprehensive experiments with real-world datasets would further strengthen the empirical validity of the presented analysis results.
>
> Our experimental design already goes beyond purely toy settings by combining **controlled hybrid real–synthetic datasets with standard CNN architectures** to test how extent and amplitude affect **which features are learned earlier**, while keeping the learning dynamics interpretable.
>
> The focus of this work is to develop a theoretically grounded understanding of when features are learned earlier or later in redundancy-reduction SSL and to validate these mechanisms in settings where extent and amplitude can be precisely manipulated. To this end, we use:
> - Controlled hybrid real–synthetic datasets (Colored MNIST, Modified Waterbirds), where object/background extent and spurious correlations can be explicitly controlled while retaining realistic image structures.
> - Standard architectures (ResNet-18/34) trained with Barlow Twins, so that our predictions are tested in familiar CNN-based pipelines.
>
> These experiments show that the theoretically predicted extent bias and eigenvalue-driven learning priority do appear in realistic architectures, including the characteristic accuracy plateau that shortens as we increase object extent.
>
> We will expand the description and analysis of these experiments in Section 7 (especially for Modified Waterbirds), making the empirical evidence more visible in the main text. In parallel, we have started extending our framework to ImageNet pretraining and downstream evaluations, and we plan to include the corresponding results in the revised manuscript during the rebuttal phase.

---

> ### Author Response · Authors · 2025-11-21
>
> > Q1 : What is the dependence of the $\Gamma$ matrix on the chosen data augmentations?
>
> $\Gamma$ is defined as an expectation over augmented pairs $(x, x′)$, so changing the data augmentation pipeline directly changes $\Gamma$.
>
> To make this dependence concrete, we added a new analysis and toy experiment on Random Resized Crop (RRC) in Appendix E.5. We focus on RRC because recent work on Joint Embedding Architectures (JEA) shows that strong performance can be achieved using essentially only random cropping, without domain-specific augmentations [1].
>
> In our toy setup, RRC acts as a spatial filter that reshapes the eigenvalues of $\Gamma$ depending on where a feature is located in the image. Features near the center tend to retain relatively large eigenvalues, while identical features moved toward the border experience an exponential suppression of their eigenvalues under more aggressive crops. Thus, even when two features have the same semantic content, peripheral features obtain smaller eigenvalues in $\Gamma$ and are learned later under redundancy-reduction SSL.
>
> Our framework therefore predicts a trade-off: making RRC less aggressive should increase background shortcuts (since background regions are more consistently preserved across views) but reduce center bias, whereas making RRC more aggressive should reduce background shortcuts but amplify center bias. Full details of this analysis are provided in Appendix E.5.
>
>
>
> > Q2 : Could you elaborate on the generalizability of the observations to other SSL algorithms? The derivations are based on the Barlow Twins loss, but can the underlying intuitions about common features of algorithms/losses to extend results to other losses or algorithmic designs?
>
> Our closed-form derivations are specific to Barlow Twins, but we **empirically observe similar eigenvalue-driven priorities in SimCLR and VICReg**, indicating that the same mechanism for which features are learned earlier likely influences a broader class of SSL objectives.
>
> - **Empirical evidence for SimCLR and VICReg.**
>   In Section 6.3 and Appendix C, SimCLR and VICReg, applied to the same toy and semi-synthetic setups, also display stepwise feature learning where dominant (large-extent or large-amplitude) features are learned earlier. Although we do not derive closed-form $\tau_j$ for these methods, the qualitative dynamics are consistent with our eigenvalue-based interpretation.
>
> Thus, we provide a detailed theory for Barlow Twins, and experiments indicate that similar eigenvalue-driven learning priorities operate in SimCLR and VICReg as well, suggesting that our framework captures a mechanism that likely influences a variety of SSL objectives.
>
>
> > Q3 : Regarding the extent vs. semantic impact: would it be possible to conduct controlled experiments with manipulated data (e.g., keeping the objects fixed while modifying the background—by expanding, changing texture, etc.) to test the influence of these properties?
>
> On **extent vs. semantic impact** and controlled manipulations, our Colored MNIST and Modified Waterbirds setups were designed precisely to disentangle spatial extent from label semantics and to test how extent alone affects **which features are learned earlier**.
>
> - **Colored MNIST (Section 7.1).**
>   We construct inputs where the semantic object (digit) is fixed, while the background color and spatial extent are controlled. By varying the object-to-background pixel ratio, we change the extent of the object feature without altering label semantics. As the object’s extent increases, the shortcut plateau shortens and the object feature is learned earlier, in line with our prediction that larger extent leads to larger eigenvalues and earlier learning.
>
> - **Modified Waterbirds (Section 7.2).**
>   In Modified Waterbirds, we manipulate background regions and correlations while keeping labels and object structure fixed. This allows us to study how changing background extent and correlations affects which feature (object vs. background) is learned earlier and how long the model stays in the shortcut regime.
>
> We have clarified in the revised Section 7 that these setups are explicitly designed to disentangle “extent” from “semantic content”: by controlling extent and background while holding labels and object identity fixed, we empirically test that spatial extent itself controls the learning order, i.e., which feature is learned earlier, independent of semantics.
>
> ---
>
> ### References
>
> [1] Moutakanni, Théo, et al. "You don’t need domain-specific data augmentations when scaling self-supervised learning." Neural Information Processing Systems, 2024.

---

> > ### Comment · Reviewer_cfsZ · 2025-11-27
> >
> > I went over the responses of the authors to reviewer questions. I would like to thank the authors for the clarifications they provide. I preserve my  positive view about the value of the insights provided by the article.

---

### Official Review · Reviewer_xbxN · 2025-10-31

**Soundness:** 3
**Presentation:** 3
**Contribution:** 3
**Rating:** 6
**Confidence:** 3

**Summary:**

This paper provides a theoretical framework analyzing shortcut learning in self-supervised learning (SSL) through the lens of eigenvalue decomposition of feature cross-correlation matrices. The authors introduce two concepts: extent bias (prioritizing features based on dimensional coverage) and amplitude bias (prioritizing features based on magnitude). Building on Simon et al. (2023)'s work on stepwise learning dynamics, they demonstrate that learning priority is fundamentally governed by eigenvalues of the feature cross-correlation matrix rather than semantic importance. The theoretical analysis is validated through toy models (linear networks, MLPs) and extended to semi-realistic datasets (Colored-MNIST, Modified Waterbirds).

**Strengths:**

Rigorous theoretical framework: The eigenvalue decomposition analysis (Theorems 4.1, 4.2, 4.3, 5.1) provides precise mathematical characterization of when features are learned, with critical time points τⱼ ∝ 1/γⱼ clearly derived.

Clear experimental validation: Figure 1 demonstrates excellent alignment between theoretical predictions (dashed lines) and empirical results (solid lines) for loss, eigenvalues, and feature alignment evolution.

Comprehensive scope: Analysis extends beyond basic Barlow Twins to multiple SSL methods (SimCLR, VICReg - Section C), network architectures (linear, DLN, MLP - Sections B.1-B.3), and the redundancy reduction coefficient λ (Section 6.1, Figure 2).

Novel formalization: Extent bias and amplitude bias provide useful conceptual frameworks. The connection between feature dimensionality (mₗ vs mₛ) and eigenvalue magnitude (γₗ = mₗ > γₛ = mₛ, Theorem 4.1) is elegantly established.

Some empirical validation: Colored-MNIST experiments (Section 7.1, Figure 5) show the plateau at 70% accuracy directly validates the extent bias hypothesis in a controlled setting with varying object ratios.

**Weaknesses:**

Limited novelty of core insight: The observation that models learn high-dimensional/high-amplitude features first is well-established in the spectral bias literature (Rahaman et al. 2019, Tancik et al. 2020 - cited by authors) and also formulated as various other names (easy-to-learn, low-variance features etc.) in the literature. The main contribution is formalizing this specifically for SSL eigenvalue dynamics.

Limited actionable insights: While the paper explains why shortcut learning occurs, it offers minimal guidance on how to mitigate it. The conclusion mentions "designing mechanisms to encourage learning of generalizable features" but provides no concrete methods.
Experimental scope:

Modified Waterbirds experiments (Section 7.2) are interesting but only briefly described in appendix.
No experiments on standard SSL benchmarks (ImageNet pretraining + downstream tasks) to assess real-world impact.
The 70% plateau observation in Figure 5 is compelling but limited to artificially constructed spurious correlations.

**Questions:**

What do the authors think the key takeaway from this work should be?

Is the goal to provide a theoretical framework for future analysis or does this yield some clear empirical insights for mitigating or discovering spurious correlations in practice already?

---

> ### Author Response · Authors · 2025-11-21
>
> We sincerely thank the reviewer for the detailed and constructive feedback. Below we address each point in turn.
>
> > W1: Limited novelty of core insight: The observation that models learn high-dimensional/high-amplitude features first is well-established in the spectral bias literature (Rahaman et al. 2019, Tancik et al. 2020 - cited by authors) and also formulated as various other names (easy-to-learn, low-variance features etc.) in the literature. The main contribution is formalizing this specifically for SSL eigenvalue dynamics.
>
> Redundancy-reduction SSL **does not follow** the same learning mechanisms as spectral bias in supervised learning. In our setting, feature learning is driven by extent- and amplitude-dependent eigenvalues of the feature cross-correlation matrix, rather than by frequency-based biases studied in supervised setups.
>
> While it is well established that neural networks exhibit learning biases, the extent/amplitude bias we analyze in SSL **is distinct from** spectral bias and frequency shortcut (L307–310):
>
> - **Spectral bias**: In **supervised** regression, frequency typically determines learning priority—*low-frequency* components are learned earlier, often regardless of their amplitude, **even though they may have much larger amplitudes**.
>
> - **Frequency shortcut**: In **supervised** classification, there are frequency shortcuts that neural networks tend to learn—the most *distinctive* frequency characteristics for easy classification, **which can be either low or high frequencies**.
>
> In contrast, we prove that in redundancy-reduction SSL, the learning order is governed by the eigenvalues $\gamma_j$ of the feature cross-correlation matrix $\Gamma$. As a consequence, features with larger extent or amplitude are learned earlier, **regardless of their frequency**.
>
> This eigenvalue-based mechanism allows us to explain why SSL models prioritize shortcut features (e.g., backgrounds with large spatial extent) even when they are high-frequency or semantically less relevant. Thus, rather than a reformulation, our work reveals that eigenvalue magnitude dictates the stepwise learning order in redundancy-reduction SSL.
>
> ---
> > Q1: What do the authors think the key takeaway from this work should be?
>
> The key takeaway of our work is that, in redundancy-reduction SSL, which features are learned earlier during training is governed by the eigenvalues of the feature cross-correlation matrix $\Gamma$, rather than by their semantic importance. Feature extent (spatial coverage) and amplitude both increase these eigenvalues, so high-extent or high-amplitude features are learned earlier, even when they correspond to shortcut or spurious patterns such as backgrounds.
>
> This eigenvalue-based view explains shortcut learning in SSL as a generic property of the objective and data statistics, and provides a quantitative link between data design, augmentations, and which features are learned earlier.

---

> ### Author Response · Authors · 2025-11-21
>
> > W2: Limited actionable insights: While the paper explains why shortcut learning occurs, it offers minimal guidance on how to mitigate it. The conclusion mentions "designing mechanisms to encourage learning of generalizable features" but provides no concrete methods.
>
> Concretely, our analysis suggests the following three practical strategies for SSL practitioners:
>
> 1. **$\lambda$-tuning (decorrelation strength)**
> Our analysis for Barlow Twins with general redundancy reduction $0 \le \lambda \le 1$ shows that appropriately increasing the redundancy reduction term mitigates the dominance of (potentially spurious) large-extent/amplitude features and facilitates learning of other more predictive features. This provides a principled guideline for setting $\lambda$ beyond heuristic tuning aimed purely at decorrelation.
>
> 2. **Data preprocessing design (object–background extent)**
>    Our Colored MNIST and Modified Waterbirds experiments demonstrate that enlarging object regions or using crops that reduce background extent shortens the shortcut-dominated plateau. This suggests a concrete data/preprocessing strategy: when possible, increase the object-to-background ratio to mitigate shortcut learning driven by background features.
>
> 3. **Data augmentation design (extent bias via RRC)**
>    Our analysis yields a simple design guideline: **by adjusting the aggressiveness of Random Resized Crop (RRC), one can regulate the trade-off between rapidly learning central targets and preserving peripheral features**. Using aggressive RRC can induce a "center bias"—a tendency where the model relies primarily on the central region of an image. This phenomenon stems from the theoretical definition of $\Gamma$ as an expectation over augmented pairs $(x, x')$; modifying the cropping pipeline directly reshapes the spectral properties of $\Gamma$. We focused on RRC, as recent work [1] on Joint Embedding Architectures (JEA) suggests that strong performance can be achieved using essentially only random cropping. As detailed in our controlled experiments (Appendix E.5), RRC acts as a spatial filter. Fig. 35 empirically demonstrates this mechanism: the maximum eigenvalue $\gamma_{max}$ monotonically decreases as a feature moves away from the image center, and this decay becomes significantly steeper under aggressive cropping (lower minimum scale $r$, e.g., 0.3), thereby suppressing the eigenvalues of peripheral features. To validate this effect on learning dynamics, we utilized a synthetic dataset containing semantically independent 'Center Circle' and 'Outer Square', allowing us to decouple spatial location from semantic content (Fig. 37). The resulting performance dynamics confirm that aggressive cropping (ratio $[0.08, 1.0]$) amplifies the center bias, rapidly minimizing error for the center object while degrading performance on the outer object. Conversely, a milder cropping strategy (ratio $[0.7, 1.0]$) mitigates this suppression, allowing the model to effectively learn the outer object. These findings yield design guidelines, suggesting that adjusting crop aggressiveness regulates the trade-off between learning central targets and peripheral features.
>
> We will revise the conclusion and discussion sections to explicitly summarize these three levers (decorrelation strength, object–background extent, and RRC aggressiveness) as concrete guidance for practitioners.

---

> ### Author Response · Authors · 2025-11-21
>
> > W3: Experimental scope: Modified Waterbirds experiments (Section 7.2) are interesting but only briefly described in appendix. No experiments on standard SSL benchmarks (ImageNet pretraining + downstream tasks) to assess real-world impact. The 70% plateau observation in Figure 5 is compelling but limited to artificially constructed spurious correlations.
>
>
> While our experimental scope is limited and does not include standard large-scale SSL benchmarks, our goal in this work is to provide a theoretical framework validated on controlled and hybrid real–synthetic datasets where we can precisely manipulate feature extent and amplitude and directly test which features are learned earlier.
>
> To bridge theory and practice, we designed hybrid real–synthetic experiments that retain realistic images and architectures while allowing us to control biases:
>
> - **Realistic architectures and hybrid real–synthetic datasets.**
>   In Sections 7.1 and 7.2, we use ResNet-18/34 on Colored MNIST and Modified Waterbirds. These setups maintain control over object/background extent and correlations while using realistic convolutional networks. The extent-driven “shortcut plateau” predicted by our theory (e.g., around 70% accuracy when the background–label correlation is 70%) consistently appears in these settings, indicating that the eigenvalue-based learning order persists beyond linear models.
>
> - **Strengthening the description of Modified Waterbirds.**
>   Following the reviewer’s comment, we moved part of the Modified Waterbirds description and analysis from the appendix into the main text (Section 7.2), including more details on dataset construction, spurious correlations, and plateau behavior. This makes the connection between extent bias and hybrid real–synthetic data more explicit.
>
> - **Extent bias beyond artificially constructed spurious correlations.**
>   To address the concern that the 70% plateau might be an artifact of artificially enforced 70% spurious correlation, we added an additional experiment on a variant of Modified Waterbirds-B (Appendix E.6) where the natural backgrounds are made independent of the labels (no spurious correlation, objects are now independent of backgrounds). In this setting there is no 70% plateau, yet we still observe that examples with larger object extent are learned faster: as we increase the bird-to-image area ratio, the number of training steps required to reach 80% accuracy monotonically decreases. This shows that the effect we emphasize—features with larger extent being learned earlier—is not tied to a particular artificial correlation level, but persists even when backgrounds carry no label signal.
>
> While extending our framework to full-scale benchmarks such as ImageNet pretraining and downstream evaluations is clearly of practical interest, such settings involve many entangled factors (data scale, architecture details, optimization tricks, and domain-specific biases), which makes it difficult to isolate and rigorously test the specific mechanisms we study. In this work, we therefore deliberately focus on controlled and hybrid real–synthetic datasets where feature extent and amplitude can be precisely manipulated and the causal link to learning order can be cleanly validated. In parallel, we have started extending our framework to ImageNet pretraining and downstream evaluations, and we plan to include the corresponding results in the revised manuscript during the rebuttal phase.
>
> ---
> > Q2: Is the goal to provide a theoretical framework for future analysis or does this yield some clear empirical insights for mitigating or discovering spurious correlations in practice already?
>
> Our primary goal is to provide a theoretical framework for stepwise feature learning in redundancy-reduction SSL, but this framework already yields empirical insights for practice as discussed in our response to W2.
>
> We will emphasize these points explicitly in the Discussion, so that the paper serves both as a theoretical foundation and as a guide to designing augmentations and datasets that encourage more robust features to be learned earlier.
>
> ---
>
> ### References
>
> [1] Moutakanni, Théo, et al. "You don’t need domain-specific data augmentations when scaling self-supervised learning." Neural Information Processing Systems, 2024.

---

### Author Response · Authors · 2025-12-03
**Revision**

We have made the following major changes:

- Added Section 8 (Practical Guidance) to the main text
- Added detailed explanation distinguishing extent bias and amplitude bias at the beginning of Section 4
- Reorganized the appendix structure for better clarity
- Added Random Resized Crop experiments in Appendix E.5 (Figures 35, 36, 37)
- Added augmented background experiments on Modified Waterbirds-B (no spurious settings, independent background/object) in Appendix E.6
- Added ImageNet experiments with ResNet-50 (25M params) in Appendix E.7 (Figure 39), confirming that images with small extent bias learn backgrounds earlier and objects later

We believe these additions address the reviewers' concerns and strengthen our paper.

---

### Author Response · Authors · 2025-12-03
**Summary Comment**

Dear AC and Reviewers

We appreciate the reviewers' constructive feedback.

Two reviewers (xbxN, cfsZ) rated the paper as “marginally above the acceptance threshold” (6), highlighting the soundness and clarity of the theoretical framework. The third reviewer (Q3Bm) rated the paper lower (2), with main concerns regarding novelty and practical significance. However, we believe the revisions and clarifications provided during the review period substantially address these concerns.

The main concerns center on **(1) novelty over Simon et al.** and **(2) distinction from other prior studies on bias**, **(3) limited Experimental Scope**, and **(4) the practical impact of our findings**. Below we summarize the strengths and how our revisions address the concerns.

---
## Strengths

### 1. Theoretical Contribution (Acknowledged by xbxN, cfsZ, Q3Bm)

- Eigenvalue decomposition provides **precise mathematical characterization** of feature learning with critical time points $\tau_j \propto 1/\gamma_j$. (xbxN)
- Extent bias and amplitude bias are **elegantly established** as useful conceptual frameworks. (xbxN)
- **Sound proofs** with analyses validated on real data. (cfsZ)
- **Valuable extension of prior research**, both extent bias and frequency (amplitude) bias have been recognized in the literature as important issues, and this paper provides insights by applying the analytical framework of Simon et al. (cfsZ)
- Connects **stepwise feature learning in SSL to shortcut learning**. (Q3Bm)

### 2. Experimental Validation (Acknowledged by xbxN, cfsZ, Q3Bm)
- **Excellent alignment** between theoretical predictions and empirical results (loss, eigenvalues, feature alignment). (xbxN)
- **Comprehensive scope**: multiple SSL methods (SimCLR, VICReg), architectures (linear, DLN, MLP), and varying λ. (xbxN)
- Colored-MNIST experiments **directly validate** the extent bias hypothesis. (xbxN)
- Synthetic experiments in **more general settings** with evaluations on semi-synthetic image datasets. (cfsZ, Q3Bm).
- Real/semi-real data (Colored-MNIST, Modified Waterbirds) validation **enhances credibility** of findings. (cfsZ)

### 3. Clarity (xbxN, cfsZ)
- Critical time points **clearly derived**, the connection between feature dimensionality and eigenvalue magnitude is **elegantly established**. (xbxN)
- Well-presented, easy and pleasant to read (cfsZ)
- **Clear problem statements** and theorems (cfsZ)

---

## Addressed Concerns

### 1. Novelty over Simon et al (Q3Bm)
- Limited novelty; largely same stepwise dynamics on two toy examples.

### 2. Distinction from Prior Bias Studies (xbxN, Q3Bm)
- Core insight is "well-established" in spectral bias literature (xbxN).
- Feature suppression analysis already appears in Xue et al. (2023). (Q3Bm)

### 3. Limited Experimental Scope (xbxN, cfsZ)
- No experiments on standard SSL benchmarks (e.g., ImageNet pretraining) (xbxN).
- Toy linear/single-layer model is "too simple." More comprehensive real-world experiments would strengthen empirical validity (cfsZ).

### 4. Lack of Actionable Insights (xbxN, Q3Bm)
- Explains *why* shortcut learning occurs but minimal guidance on *how to mitigate* it (xbxN).
- Unclear what new practical guidance can be drawn (Q3Bm).
---

### Summary

Our work makes distinct contributions beyond prior art:
1. **Novelty**: Identifies *what* determines learning order in SSL (eigenvalues of $Γ$), not just *that* stepwise learning occurs, with theoretical analysis across linear, deep linear, and MLP models.
2. **Distinction**: Extent/amplitude bias differs from spectral bias (Rahaman et al.), frequency shortcuts (Geirhos et al.), and feature suppression in contrastive learning (Xue et al.); arises only in redundancy-reduction SSL
3. **Actionable insights**: Three concrete mitigation strategies ($λ$-tuning, preprocessing, RRC adjustment)
4. **Scalability**: Validated from toy models to ResNet-50 (params: 25M), from synthetic to ImageNet datasets

We believe these contributions address the reviewers' concerns and demonstrate both theoretical depth and practical relevance. Below, we provide detailed responses to the four main concerns: (1) novelty over Simon et al., (2) distinction from prior bias studies, (3) limited experimental scope, and (4) lack of actionable insights.

---

> ### Author Response · Authors · 2025-12-03
>
> ## 1. Novelty over Simon et al. (xbxN)
> While Simon et al. (2023) established *that* SSL exhibits stepwise learning, our work addresses: **which features are learned first, how realistic loss dynamics behave, and how these dynamics connect to shortcut learning**. The key differences are summarized below:
> |  | Simon et al. (2023) | **Ours** |
> |-|-|-|
> | **Core Question** | **Why** does SSL exhibit stepwise learning? (Focus on existence) | **Which** features are learned first? (Focus on determinants: Extent & Amplitude) |
> | **Feature Semantics** | Eigenvectors remain mathematical "directions" **without semantic** interpretation. | Eigenvectors are **linked to physical structures** (Background vs. Object, Low vs. High Freq). |
> | **Loss Dynamics** | Analyzed **only linear Barlow Twins with $λ=1$** (independent learning). | Analyzed **general redundancy reduction term ($0≤λ≤1$)** (Section 6.1, A.5). |
> | **Architecture Scope** | Primarily **Linear** models. | Verified across **Linear** (Figs. 6, 7), **Deep Linear** (Figs. 20, 21), and **MLPs** (Figs. 22, 23). |
> | **Implication** | Theoretical observation of learning phases. | **Causal link to Shortcut Learning,** naturally suggesting mitigation strategies ($λ$-tuning, preprocessing, augmentation). |
>
> ---
> ## 2. Distinction from bias studies
> ### 2.1 Conceptual Clarity (Q3Bm)
> Extent and amplitude are distinct axes on images: spatial coverage vs. per-pixel intensity.
> - Extent bias: eigenvalue scales with dimensionality ($γ=m$, Theorem 4.1)
> - Amplitude bias: eigenvalue scales with signal magnitude ($γ∝A^2$, Theorem 5.1)
> ### 2.2 "Larger Features Learned First" is not **obvious** (xbxN, Q3Bm)
> | Work | Finding | Relation to Magnitude |
> |-|-|-|
> | Rahaman et al. (2019), Basri et al. (2019) | Low frequencies learned first (supervised regression) | Almost **independent of amplitude** |
> | Wang et al. (2023) | Networks learn "distinctive" frequencies (supervised classification) | Can be **high or low frequencies** |
> | Geirhos et al. (2018) | CNNs prefer local texture over global shape (supervised classification) | Almost **independent of magnitude** |
>
> **Conclusion**: The extent and amplitude biases are specific to **redundancy-reduction SSL**, not a general property of neural networks.
> ### 2.3 Fundamentally **different settings** from Xue et al. (Q3Bm)
> | | Xue et al. (ICML 2023) | Ours |
> | - | - | - |
> | **Framework** | **Supervised / Unsupervised Contrastive Learning** with positive/negative pairs  | **Redundancy Reduction SSL** with cross-correlation matching (no explicit negatives) |
> | **Mechanism** | **SGD implicit bias** toward min-norm: "easy" features dominate, "hard" features suppressed/unlearned | Eigenvalue decomposition: larger eigenvalues → faster learning, features co-exist |
> | **Key Phenomenon** | **Feature Collapse**: permanent suppression of subclass-relevant features | **Stepwise Dynamics**: learning priority issue, not permanent suppression |
>
> Xue et al. explain *feature collapse* under **contrastive learning**; we explain *learning priority* in **redundancy-reduction SSL**.
>
> ---
> ## 3. Expanded Experimental Coverage (xbxN, cfsZ)
> We aim for a testable theoretical framework under controlled yet realistic conditions, rather than exhaustive large-scale benchmarking.
> ### 3.1 Hybrid real–synthetic settings with realistic architectures
> - **Realistic architectures and hybrid datasets.**
> We use ResNet-18/34 on Colored MNIST and Modified Waterbirds (Sections 7.1, 7.2). With spurious correlations, we observe shortcut plateaus that shorten as our theory predicts.
> - **Extent bias beyond artificially constructed spurious correlations.** (new)
> On a variant of Modified Waterbirds-B with label-independent backgrounds, learning time decreases as object ratio increases—confirming extent bias persists without spurious correlations.
> ### 3.2 Toward large-scale SSL benchmarks (ImageNet, new)
> Training ResNet-50 SSL on ImageNet, we observe that classes with smaller-extent objects are learned more slowly (Fig. 39), confirming our mechanism extends to large-scale settings.
>
> ---
> ## 4. Practical Guidelines  (cfsZ, Q3Bm)
> Our analysis leads to three concrete design levers.
> ### 4.1 $λ$-tuning (decorrelation strength)
> Analysis for $0≤λ≤1$ (Section 6.1, A.5) shows that increasing $λ$ suppresses dominant shortcuts and accelerates learning of smaller-extent/amplitude features.
> ### 4.2 Data preprocessing design (object–background extent)
> Design high object-to-background ratio inputs to reduce shortcut dominance. Experiments on Colored MNIST and Modified Waterbirds confirm enlarging the object region shortens the shortcut plateau.
> ### 4.3 Data augmentation design (extent bias via RRC, new)
> RRC most directly modulates extent bias by acting as a spatial filter—eigenvalues associated with peripheral features decay more steeply under aggressive cropping (Fig. 35). Using a synthetic dataset (Fig. 37), aggressive cropping induces center bias, while milder cropping preserves peripheral features.

---

### Meta-Review · Area_Chair_FrnJ · 2026-01-14

**Summary:**

Reviewers agreed that the paper is clearly written and presents a technically sound analysis of stepwise learning dynamics in redundancy-reduction SSL, with toy-theory predictions that qualitatively align with controlled experiments. The decision-relevant concerns were whether the contribution is meaningfully distinct from prior stepwise-learning analyses (notably Simon et al. (2023)) and related work on learning biases, and whether the paper demonstrates practical impact beyond illustrating a plausible mechanism. Two reviewers near the acceptance threshold noted that the core message may be familiar in spirit, that the analysis relies on highly simplified constructions, and that the initial empirical validation did not include standard large-scale SSL evaluations or downstream impact. The third reviewer argued that the extent/amplitude framing is not sufficiently separated from magnitude effects in the toy examples, that novelty over prior work (including feature dominance or suppression analyses) is limited, and that the practical implications remain unclear. Overall, the discussion focused on novelty, generality beyond toy settings, and actionable value.

**Reviewer Concerns:**

The rebuttal addressed several presentation and scope issues by clarifying the intended conceptual distinction between extent and amplitude, explaining more directly how the cross-correlation matrix depends on data augmentations, and articulating concrete levers such as tuning the redundancy-reduction strength, manipulating object-to-background extent, and adjusting crop aggressiveness. The added Random Resized Crop analysis makes the augmentation dependence more tangible, and the added ImageNet/ResNet-50 analysis and additional experiments in more realistic architectures partially address the concern that the work is overly toy. However, key concerns remain. The novelty critique is only partially resolved: while the authors position the contribution as identifying determinants of learning order via eigenvalues and mapping eigenmodes to interpretable structures, the simplified constructions can make the eigenvalue ordering closely track the chosen notion of extent or amplitude, and the conceptual distance to existing "easy features first" narratives is not fully established. The practical significance also remains limited: the proposed levers are plausible, but the evidence does not yet demonstrate that they reliably translate into improved robustness or downstream performance in standard SSL evaluation pipelines. The new ImageNet evidence is suggestive but primarily correlational and does not substitute for end-to-end benchmark validation of the actionable guidance.

**Reviewer Scores:**

Reviewer xbxN provided a borderline-accept score and emphasized limited novelty relative to related analyses and limited evaluation on standard large-scale SSL settings, along with limited actionable guidance. The rebuttal improves framing and adds experiments, which could reasonably support a small upward revision, although the remaining gap in benchmark-style impact could also justify no change. Reviewer cfsZ similarly provided a borderline-accept score, with primary concerns about the simplicity of the analytical setting and the need for broader real-world experimental validation. The rebuttal’s added experiments and clarifications could support a modest increase, though the evidence may also be viewed as insufficient to warrant changing the original score. Reviewer Q3Bm provided a reject score, citing limited novelty, conceptual ambiguity in separating extent from amplitude in the toy setups, and unclear practical implications beyond confirming stepwise behavior. The rebuttal improves clarity and positioning relative to prior work, which could plausibly raise the score modestly, but the core objections appear only partially addressed.

---

### Decision · Program_Chairs · 2026-01-26

Reject